# DA-Raf synergistically binds to the plasma membrane and Ras to suppress ERK signaling

Kazunori Takano[1], Kazuya Tsujita[2,3], Akiko Suganami[4], Takuhiko Nakamura[1], Emiri Kanno[1], Yutaka Tamura[4], Toshiki Itoh[2,3], Takeshi Endo[1]

The small GTPase Ras on the plasma membrane (PM) activates the ERK pathway (Raf–MEK–ERK signaling pathway) to regulate a variety of cellular, physiological, and pathological events. DA-Raf1 (DA-Raf) is a splicing isoform of A-Raf and contains the Ras-binding domain and the Cys-rich domain but lacks the conserved region 2 (CR2) and CR3 containing the kinase domain. Accordingly, DA-Raf dominant-negatively regulates Raf proteins to prevent the Ras–ERK pathway. We elucidate here the mechanisms of how DA-Raf conducts its dominant-negative function on Raf proteins. Because DA-Raf lacks the CR2 and CR3, it was incapable of adopting the autoinhibitory closed conformation and thereby favorable for PM localization. Basic amino acids in DA-Raf Ras-binding domain, and those in the Cys-rich domain, were essential for the interaction with phosphatidylserine in the PM. This interaction favored the cooperative binding of DA-Raf to active Ras, which predominated over that of Raf proteins, leading to the stable PM association of DA-Raf. Consequently, DA-Raf exerts its dominant-negative function on Raf proteins to prevent the Ras–ERK pathway.

## Introduction

Extracellular signals through their receptors activate the small GTPase classical Ras (H-Ras, K-Ras, and N-Ras). Activated Ras in turn acts on its multiple effector proteins, including Raf, phosphatidylinositol 3-kinase (PI3K), and RalGEFs, to conduct a variety of cellular and physiological functions (Karnoub & Weinberg, 2008; Cox & Der, 2010). The activation of the Raf family of Ser/Thr kinases (A-Raf, B-Raf, and C-Raf) leads to the ERK pathway (Raf–MEK–ERK signaling pathway), which is one of the MAPK pathways. The Ras-activated ERK pathway regulates many fundamental cellular processes, including cell proliferation, growth, differentiation, survival, apoptosis, migration, and metabolism (McCubrey et al, 2007; Lavoie et al, 2020).

The Raf proteins share three conserved regions: CR1, CR2, and CR3 (see Fig 2A) (Matallanas et al, 2011; Lavoie & Therrien, 2015; Terrell & Morrison, 2019). The N-terminal CR1 contains a Ras-binding domain (RBD) and a Cys-rich domain (CRD), and the central CR2 includes a Ser/Thr-rich stretch. The C-terminal CR3 represents a Ser/Thr kinase domain. When Ras is inactive, B-Raf and C-Raf are phosphorylated on Ser residues in both the CR2 and the C-terminal tail after the kinase domain. They fold into autoinhibitory closed conformations via 14-3-3 protein dimer binding to the phospho-Ser residues. In the autoinhibitory closed conformation of B/C-Raf, the CRD is sequestered in the 14-3-3 dimer and cannot interact with the plasma membrane (PM), although the RBD is partially exposed (Kondo et al, 2019; Park et al, 2019; Tran et al, 2021). Accordingly, these inactive Raf proteins are diffusely present in the cytosol.

The initial essential step in Raf activation is the recruitment of Raf to the PM, which is triggered by Ras activation (Matallanas et al, 2011; Lavoie & Therrien, 2015; Terrell & Morrison, 2019; Spencer-Smith & Morrison, 2024). When PM-anchored Ras is activated by its GTP-loading, Raf binds to the activated Ras via the RBD. The Ras-binding elicits CR2 dephosphorylation and 14-3-3 release from the CR2, thereby removing Raf autoinhibition. Then, the Raf CRD interacts with phosphatidylserine (PS) in the PM, which is critical for the direct association of Raf with the PM. The C-Raf CRD also interacts with H-Ras C-terminal farnesyl groups anchored to the PM and with K-Ras directly (Thapar et al, 2004; Tran et al, 2021). Thus, both the RBD and CRD interact with Ras, albeit with distinct sites. Subsequently, Raf is activated by dimerization through 14-3-3 dimer binding to the C-terminal tail phospho-Ser residues. Raf activation also requires multiple phosphorylation of the negative charge regulatory region (N-region) and the kinase domain.

We have identified DA-Raf1 (DA-Raf), which is generated by alternative splicing of Araf pre-mRNA in vertebrates (Yokoyama et al, 2007; Endo, 2020). DA-Raf shares with A-Raf the N-terminal portion containing the RBD and CRD but lacks the CR2 and the CR3 containing the kinase domain. The domain structure of DA-Raf suggests that

[1]Department of Biology, Graduate School of Science, Chiba University, Chiba, Japan  [2]Biosignal Research Center, Kobe University, Kobe, Japan  [3]Department of Biochemistry and Molecular Biology, Kobe University Graduate School of Medicine, Kobe, Japan  [4]Department of Bioinformatics, Graduate School of Medicine, Chiba University, Chiba, Japan

Correspondence: ktakano@faculty.chiba-u.jp

DA-Raf easily associates with the PM and active Ras and antagonizes the Ras–ERK pathway in a dominant-negative manner. Indeed, DA-Raf binds to active Ras to disrupt the Ras–Raf interaction, thereby preventing MEK and ERK activation (Yokoyama et al, 2007; Watanabe-Takano et al, 2014). Endogenous DA-Raf expression is prominently induced during the differentiation of mouse skeletal myocytes, and DA-Raf serves as a master inducer of mammalian skeletal myocyte differentiation by blocking the Ras–ERK pathway (Yokoyama et al, 2007; Takahashi et al, 2019; Endo, 2023). Moreover, DA-Raf counteracts skeletal myocyte differentiation inhibition by myostatin and GDF11, which are involved in muscle atrophy and sarcopenia by preventing their non-Smad Ras–ERK pathway (Masuzawa et al, 2022). Investigations using DA-Raf knockout mice have revealed that endogenous DA-Raf is postnatally highly induced in lung alveolar epithelial type 2 cells to prevent the Ras–ERK pathway. Accordingly, DA-Raf participates in alveolar septum formation through myofibroblast differentiation in a non-cell-autonomous fashion (Watanabe-Takano et al, 2014). In addition, DA-Raf is essential for TGF-β1-induced epithelial-mesenchymal transition (EMT) in alveolar epithelial type 2 cells to myofibroblasts by interrupting TGF-β1–activated Ras–ERK pathway (Watanabe-Takano et al, 2015).

Stable expression of DA-Raf in oncogenic K-Ras-transformed fibroblasts interferes with all the K-Ras-transformed phenotypes, including tumorigenicity in a mouse xenograft model (Yokoyama et al, 2007; Kanno et al, 2018). DA-Raf with the single-nucleotide polymorphism (SNP) R52Q and DA-Raf R52W mutant detected in human lung cancer, and an R52L mutant, are incompetent to suppress the K-Ras-induced transformation. Furthermore, DA-Raf expression is silenced in KRAS-mutant human cancer cell lines. Therefore, DA-Raf can be determined as a tumor suppressor protein that targets mutant Ras-induced tumorigenesis (Kanno et al, 2018). Moreover, DA-Raf expression in the cancer cells impairs their migration and invasion abilities. Thus, DA-Raf may also function as an invasion suppressor protein in the KRAS-mutant cancer cells (Matsuda et al, 2024).

We have previously elucidated the gross mechanisms of the dominant-negative function of DA-Raf. However, because the detailed mechanisms, particularly in relation to its interaction with the PM, remained unsolved, we addressed these issues in this study. Because DA-Raf lacks the CR2 and CR3, it was incapable of adopting the autoinhibitory closed conformation, and thus, its RBD and CRD were constitutively exposed, thereby favoring PM localization. A basic amino acid cluster in the RBD was essential for interacting with PS in the PM. This interaction favored the cooperative binding of DA-Raf to active Ras, which predominated over the binding of Raf proteins to Ras. Consequently, DA-Raf exerts a dominant-negative function on Raf proteins to prevent the ERK pathway.

## Results

### DA-Raf is predominantly localized to the PM and more efficiently binds to K-Ras than do Raf proteins

Because DA-Raf acts as a dominant-negative antagonist of the Ras–ERK pathway, DA-Raf should predominate over Raf proteins in

the binding to active Ras either qualitatively or quantitatively. Thus, we first analyzed the degree of binding of EGFP-tagged Raf proteins (A-Raf, B-Raf, and C-Raf) and DA-Raf to GST-tagged, GTPγS-loaded active K-Ras by a pull-down assay. Each Raf protein and DA-Raf bound to K-Ras–GTPγS to a similar degree (Fig 1A). Thus, the binding affinity of the Raf proteins and DA-Raf for active K-Ras may be comparable in vitro.

We then analyzed the localization of the Raf proteins and DA-Raf in comparison with active K-Ras to determine how Raf proteins and DA-Raf are localized to the PM for their binding to active Ras. When EGFP-tagged Raf proteins were expressed in MDCK epithelial cells, all of them were diffusely distributed throughout the cytoplasm as detected by confocal microscopy. In contrast, EGFP–DA-Raf was predominantly located to the PM (Fig 1B and E). When TagBFP-tagged K-Ras(G12V), a constitutively active mutant of K-Ras that is essentially located to the PM, was coexpressed with Raf proteins, the Raf proteins were mobilized to the PM at high levels, but subsets of them remained in the cytoplasm (Fig 1C and E). DA-Raf remained at the PM irrespective of the K-Ras(G12V) expression (Fig 1C and E). These results were also similar in human cervical adenocarcinoma HeLa cells without the K-Ras(G12V) expression (Fig S1A and C) and those with the K-Ras(G12V) expression (Fig S1B and C). Coexpression of K-Ras(S17N), a dominant-negative K-Ras mutant, with Raf proteins resulted in the cytoplasmic diffuse distribution of Raf proteins (Fig 1D and E). On the other hand, DA-Raf was significantly restricted to the PM and also partially present in the cytoplasm upon K-Ras(S17N) coexpression (Fig 1D and E). Therefore, DA-Raf is substantially localized to the PM regardless of Ras activity, whereas the PM localization of Raf proteins critically depends on Ras activity.

We further examined whether the Raf proteins and DA-Raf directly bound to active K-Ras(G12V) at the PM also in cells by applying bimolecular fluorescent complementation (BiFC) analysis (Hu et al, 2002) with Kusabira-Green N-terminus (KGN)-tagged K-Ras(G12V) and Kusabira-Green C-terminus (KGC)–tagged Raf proteins/DA-Raf (Fig 1F). Coexpression of KGN–K-Ras(G12V) and KGC as a control did not show any fluorescence in MDCK cells, whereas coexpression of KGN–K-Ras(G12V) and KGC–A/B/C-Raf displayed fluorescence on the PM (Fig 1G and H). When KGC–DA-Raf was coexpressed with KGN–K-Ras(G12V), much stronger fluorescence was detected on the PM (Fig 1G and H). These results imply that the Raf proteins and DA-Raf directly bind to active K-Ras(G12V) at the PM. Furthermore, DA-Raf binds to active K-Ras much more efficiently than do Raf proteins in cells.

### DA-Raf can efficiently bind to active K-Ras at the PM because of the absence of the CR2

We next addressed the mechanisms of how DA-Raf is predominantly localized to the PM regardless of Ras activity. Raf proteins share three conserved regions: CR1, CR2, and CR3, whereas DA-Raf includes the CR1 but lacks the CR2 and CR3 (Fig 2A). The CR2 and CR3 contain phosphorylatable Ser residues, which are involved in the autoinhibitory closed conformation of Raf via 14-3-3 binding. The inactive Raf proteins are diffusely distributed in the cytosol without binding to Ras (Matallanas et al, 2011; Lavoie & Therrien,

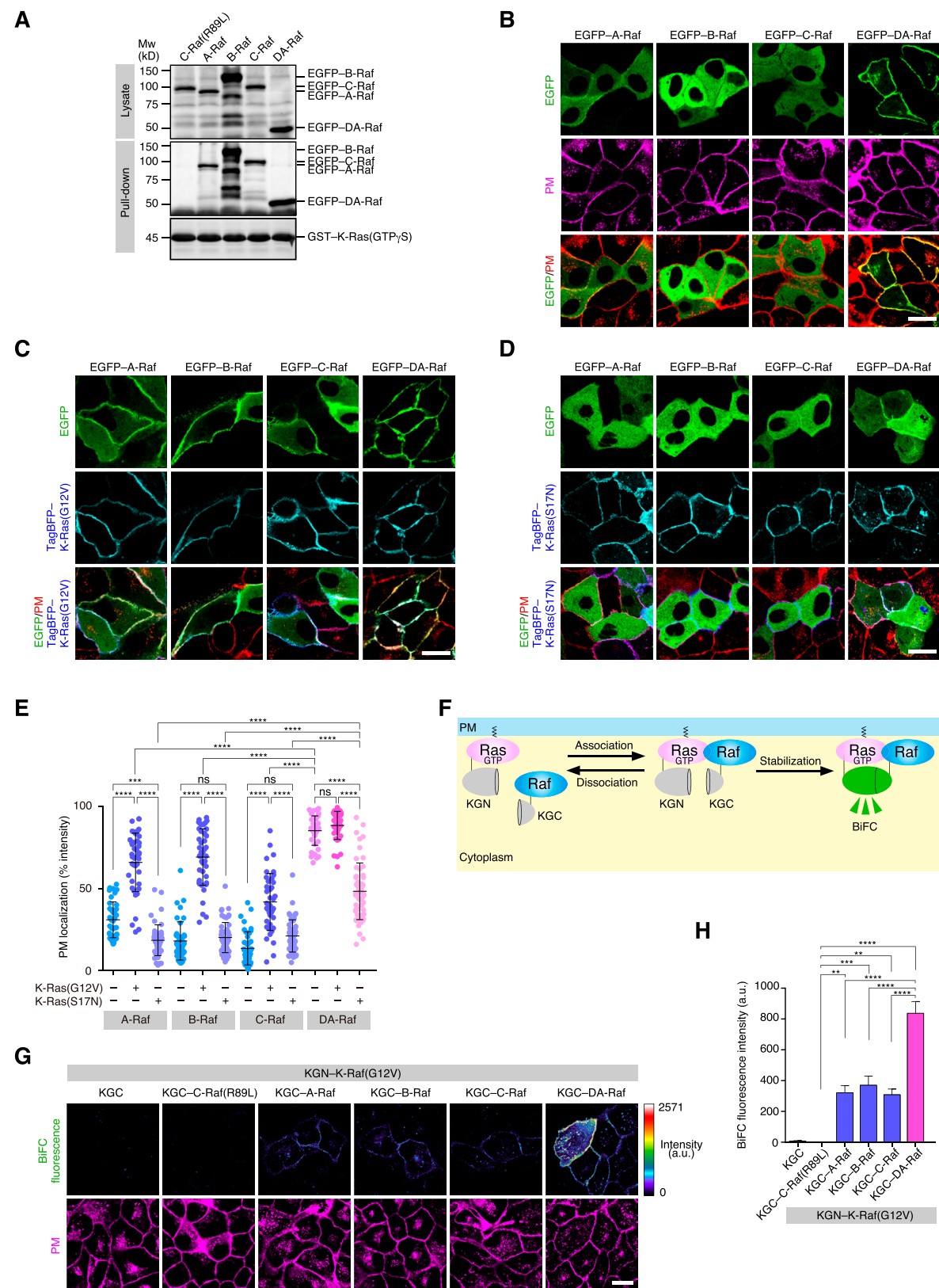

**Figure 1. Predominant localization of DA-Raf to the PM over Raf proteins and its efficient binding to K-Ras.**
**(A)** In vitro binding of Raf proteins and DA-Raf to K-Ras analyzed by a pull-down assay. Lysates of HeLa cells transfected with EGFP–A/B/C/DA-Raf or EGFP–C-Raf(R89L) were used. The binding of EGFP–A/B/C/DA-Raf to GST–K-Ras–GTPγS was detected by immunoblotting with the anti-GFP pAb. **(B)** Localization of Raf proteins and DA-Raf in MDCK cells. MDCK cells were transfected with EGFP–A/B/C/DA-Raf, and the PM was stained with CellMask Orange Plasma Membrane Stain. Shown are the

2015; Terrell & Morrison, 2019). Thus, we examined the interaction of Raf proteins and DA-Raf with 14-3-3β, a Raf-interacting 14-3-3 isoform. A pull-down assay showed that each Raf protein, but not DA-Raf, bound to 14-3-3β in vitro (Fig 2B). BiFC analysis also revealed that 14-3-3β clearly bound to A/B/C-Raf throughout the cytoplasm in MDCK cells. In contrast, the interaction between 14-3-3β and DA-Raf was hardly detected (Fig 2C and D).

Moreover, to examine the role of 14-3-3 binding to CR2 in the PM localization and the Ras binding, we analyzed the behavior of unphosphorylatable CR2 mutants of Raf proteins. EGFP-tagged mouse A-Raf(S214A), mouse B-Raf(S348A), and human C-Raf(S259A) were diffusely distributed throughout the cytoplasm as their WT proteins, although A-Raf(S214A) was partially localized also to the PM (Fig 2E and G). When these mutants were coexpressed with K-Ras(G12V), they were mobilized to the PM at high levels as their WT proteins coexpressed with K-Ras(G12V) (Fig 2F and G). A pull-down assay showed that these Raf protein CR2 mutants bound to active K-Ras–GTPγS in vitro to a degree comparable to their WT proteins and DA-Raf (Fig 2H). BiFC analysis also displayed that these CR2 mutants and DA-Raf bound firmly to K-Ras(G12V) on the PM in cells (Fig 2I and J). These results indicate that the unphosphorylated state of CR2 without the 14-3-3 binding is necessary but insufficient for Raf proteins to be localized to the PM. Their binding to active Ras is indispensable for their PM localization. This notion is consistent with the established concept of Raf localization to the PM (Matallanas et al, 2011; Lavoie & Therrien, 2015; Terrell & Morrison, 2019).

We further analyzed the stability of DA-Raf and Raf proteins and their CR2 mutants on the PM with K-Ras(G12V) by FRAP. K-Ras is stably anchored to the PM via its C-terminal farnesyl group modification. Thus, the recovery by K-Ras(G12V) turnover at the PM after photo-bleaching was slow (Fig 3A and E). In contrast, that of DA-Raf was very fast. When DA-Raf was coexpressed with K-Ras(G12V), however, the recovery of DA-Raf became as slow as that of K-Ras(G12V) itself (Fig 3A and E). On the other hand, the recovery of A-Raf (Fig 3B and E), B-Raf (Fig 3C and E), or C-Raf (Fig 3D and E) coexpressed with K-Ras(G12V) was fast but less than that of DA-Raf itself (Fig 3A and E). However, the recovery of any of the Raf protein CR2 mutants coexpressed with K-Ras(G12V) was as slow as that of K-Ras(G12V) itself or that of DA-Raf coexpressed with K-Ras(G12V) (Fig 3A–E). Together, these results imply that DA-Raf can bind to active K-Ras at the PM much more efficiently than Raf proteins, owing to the absence of the CR2, which is responsible for autoinhibition. The efficient binding of DA-Raf to active K-Ras results in its stable localization to the PM, and thereby, DA-Raf can exert its dominant-negative effect on Raf proteins.

## DA-Raf interacts with PM phosphatidylserine via the basic amino acid cluster in the RBD

Next, we focused on the mechanisms of how DA-Raf associates with the PM for the efficient binding of DA-Raf to active K-Ras. According to the crystal structure of the RBD of A-Raf/DA-Raf (Zhao et al, 2005), its surface amino acids contain two basic amino acid (bAA) clusters, that is, cluster 1 (K22, K28, and R30) and cluster 2 (K66, R68, and K69) (Fig 4A). On the other hand, R52 is located on the opposite side of the structure from these bAA clusters. This R52 in A-Raf/DA-Raf is conserved among Raf proteins and corresponds to C-Raf R89, which is an electrostatic binding partner of D38 in Ras (Fabian et al, 1994). In addition, the cluster 1 bAAs (K22, K28, and R30) in DA-Raf correspond to C-Raf R59, K65, and R67, respectively, which also participate in the binding to Ras (Kiel et al, 2004). In support of this notion, the prediction of the 3D structure of DA-Raf and K-Ras–GTP complex with AlphaFold3 (Abramson et al, 2024) showed that DA-Raf R52 and cluster 1 bAAs are located near the K-Ras-binding interface (Fig 4B). In contrast, DA-Raf cluster 2 bAAs (K66, R68, and K69) are distant from the K-Ras-binding interface (Fig 4B). Thus, the roles of the cluster 2 bAAs (K66, R68, and K69) in DA-Raf and their corresponding amino acids in Raf proteins remain obscure. Therefore, we examined the function of DA-Raf cluster 2 bAAs by replacing these bAAs with the acidic amino acid Glu. The EGFP-tagged single mutants, DA-Raf(K66E) and DA-Raf(R68E), were mainly localized to the PM, as was WT DA-Raf (Fig 4C and D). Although DA-Raf(R69E) and the double mutant DA-Raf(K66E/R68E) [DA-Raf(2E)] were also localized to the PM, they were present in the cytoplasm to some degree. In contrast, the triple mutant DA-Raf(K66E/R68E/K69E) [DA-Raf(3E)] was almost diffusely distributed in the cytoplasm (Fig 4C and D). These results suggest that the cluster 2 bAAs are cooperatively involved in RBD-mediated mobilization of DA-Raf to the PM.

We further investigated the mechanism of how the cluster 2 bAAs in DA-Raf RBD participate in the localization of DA-Raf to the PM. We analyzed the interaction between DA-Raf and phosphatidylcholine (PC) or phosphatidylserine (PS), both of which are the major phospholipids of the PM, by molecular dynamics (MD) simulations. The RBD crystal structure (Fig S2C), placed near the 100% dioleoyl-PC (DOPC) (Fig S2A) or dioleoyl-PS (DOPS) bilayer (Fig S2B) surface, was used as the initial structure for the analysis (Fig S2D). MD simulations showed that the cluster 2 bAAs and DOPC bilayer were more separated and formed no hydrogen bonds between them (Fig 4E left, Video 1). On the other hand, K66 and R68 in the cluster 2 bAAs and DOPS bilayer were close and formed hydrogen bonds between them (Fig 4E right, Video 2). However, K69 remained oriented toward the adjacent β-sheet rather than approaching PS during the simulation process, suggesting that

distributions of the EGFP-tagged proteins in comparison with the PM. Scale bar, 10 μm. **(C, D)** Localization of Raf proteins and DA-Raf in MDCK cells expressing K-Ras(G12V) (C) or K-Ras(S17N) (D). Shown are the distributions of EGFP–A/B/C/DA-Raf and TagBFP–K-Ras(G12V)/K-Ras(S17N) in comparison with the PM. Scale bar, 10 μm. **(B, C, D, E)** The PM localization ratio of Raf proteins and DA-Raf in the analyses of (B, C, D). The PM localization ratio is indicated as % EGFP fluorescence intensity on the PM in each cell. The horizontal lines indicate means ± SD. **(F)** The scheme of BiFC analysis to detect Ras–Raf binding at the PM. When the association of KGN–Ras and KGC–Raf is stabilized, Kusabira-Green fluorescence is generated. **(G)** In vivo binding of Raf proteins and DA-Raf to K-Ras(G12V) analyzed by BiFC analysis. The binding between KGN–K-Ras(G12V) and KGC–A/B/C/DA-Raf or KGC–C-Raf(R89L) in MDCK cells was detected by BiFC. The BiFC fluorescence intensity is shown in rainbow colors. The PM was stained with CellMask Orange Plasma Membrane Stain. Scale bar, 10 μm. **(G, H)** BiFC fluorescence intensity in the analysis of (G). The values are means ± SEM.

Source data are available for this figure.

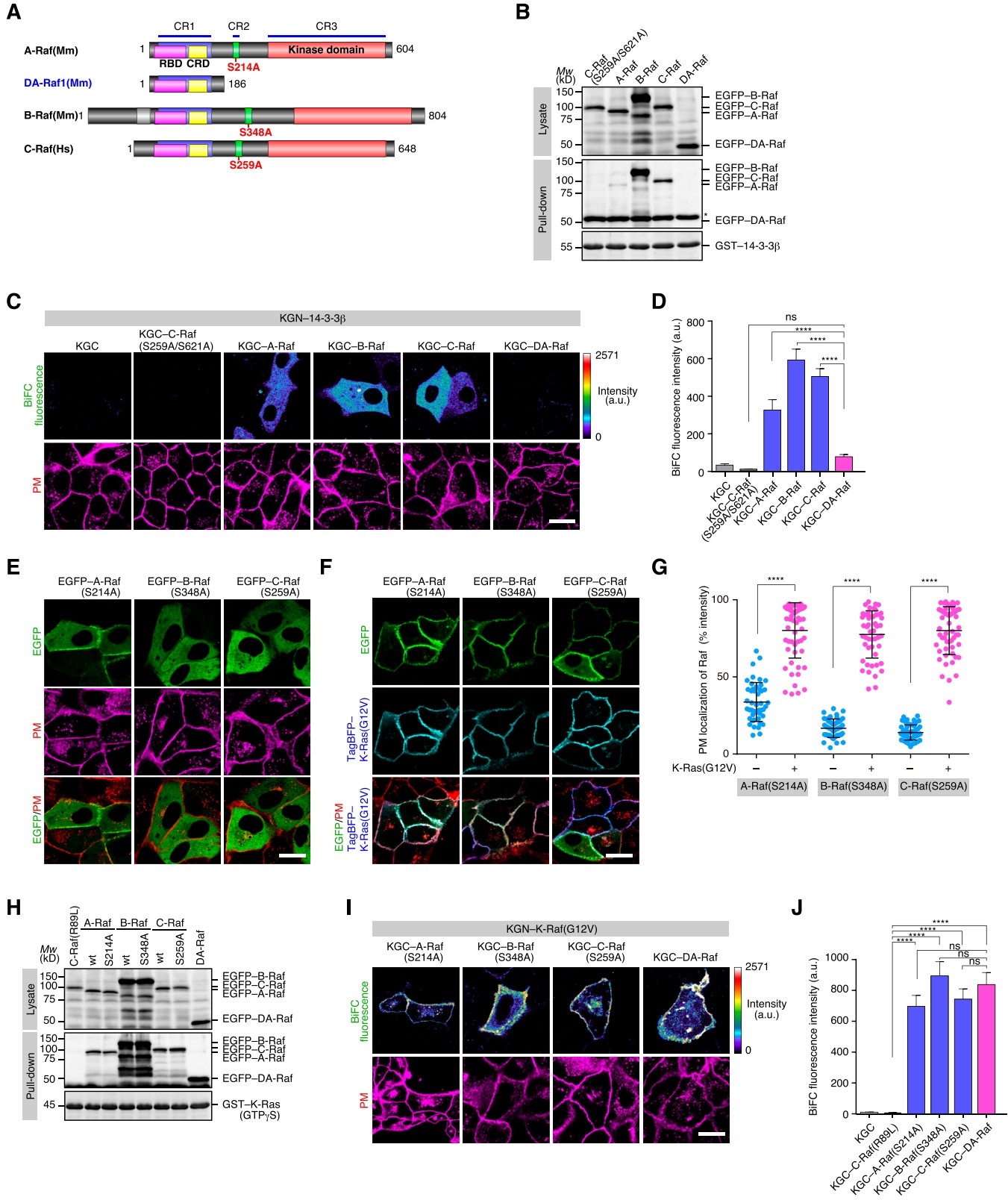

**Figure 2. Efficient binding of the unphosphorylatable CR2 mutants of Raf proteins to active K-Ras at the PM.**
**(A)** The domain structure of Raf proteins and DA-Raf. Unphosphorylatable CR2 mutations of S214A in A-Raf, S348A in B-Raf, and S259A in C-Raf are shown in red. **(B)** In vitro binding of Raf proteins and DA-Raf to 14-3-3β analyzed by a pull-down assay. The binding of EGFP–A/B/C/DA-Raf and EGFP–C-Raf(S259A/S621A) in HeLa cell lysates to GST–14-3-3β was detected by immunoblotting. **(C)** In vivo binding of Raf proteins and DA-Raf to 14-3-3β analyzed by BiFC analysis. The binding between KGN–14-3-3β and KGC–A/B/C/DA-Raf in MDCK cells was detected by BiFC. KGC–C-Raf(S259A/S621A) was used as a negative control. The PM staining is also shown. Scale bar, 10 μm.

K69 is involved in an intramolecular interaction. Moreover, the interaction energy of the binding between DA-Raf RBD and the DOPS bilayer during MD simulations was much lower than that of the association between DA-Raf RBD and the DOPC bilayer (Fig 4F). These results suggest that the interaction of K66 and R68 in the cluster 2 bAAs with PS in the PM and the intramolecular interaction within RBD by K69 are responsible for the PM localization of DA-Raf.

We then explored the in vitro interaction between DA-Raf and liposomes composed of phospholipids by a liposome-binding assay. GST-tagged DA-Raf was coprecipitated with the PS-containing liposomes derived from Folch's fraction, although GST–DA-Raf(3E) and GST itself were scarcely coprecipitated with the liposomes (Fig 4G and H). Similarly, GST–DA-Raf was coprecipitated with liposomes composed of phosphatidylethanolamine (PE)/PC/PS (PE:PC:PS = 4:4:2, a near-physiological ratio), whereas it was hardly coprecipitated with PE/PC liposomes without PS (Fig 4I and J). In addition, GST–DA-Raf(3E) was coprecipitated with the PE/PC/PS liposomes at a low level (Fig 4I and J). These results imply that the cluster 2 bAAs are essential for the interaction with PS present in the inner leaflet of the PM in a charge-dependent manner. Therefore, the interaction of the cluster 2 bAAs with PS is responsible for the PM localization of DA-Raf independently of Ras activation.

It is widely accepted that the Raf CRD interacts with the PS in the PM, which is critical for the direct association of Raf with the PM (Lavoie & Therrien, 2015; Spencer-Smith & Morrison, 2024). Thus, we further assessed whether and how much the DA-Raf CRD participates in the interaction with PS. GST-tagged DA-Raf CRD mutant (CRDm) (R103E/K104E/K117E), which corresponds to human C-Raf CRD mutant (R143E/K144E/K157E), was coprecipitated with PE/PC/PS liposomes at a lower level than was GST–DA-Raf (Fig 4I and J). The level was not significantly different from that of GST–DA-Raf(3E), although the former [DA-Raf(CRDm)] level tended to be higher than the latter [DA-Raf(3E)] level (Fig 4I and J). Consequently, the DA-Raf CRD, as well as the cluster 2 bAAs in the RBD, may also play a crucial role in the interaction with PS.

### PM localization of DA-Raf is cooperatively regulated by the interaction with PS in the PM and the binding to active Ras

We next examined the role of the cluster 2 bAAs in DA-Raf RBD and DA-Raf CRD in the binding of DA-Raf to Ras on the PM. A pull-down assay showed that DA-Raf(3E) bound to K-Ras–GTPγS at a level comparable to that of wt DA-Raf and that DA-Raf(CRDm) did at a lower level (Fig 5A). In contrast, DA-Raf(R52L), which is a mutant corresponding to C-Raf(R89L) (Fabian et al, 1994) and incapable of binding to active Ras (Kanno et al, 2018), did not bind

to K-Ras–GTPγS (Fig 5A). Microscopic observations showed that EGFP-tagged wt DA-Raf was localized to the PM, regardless of the coexpression of K-Ras(G12V) (Fig 5B and C). On the other hand, DA-Raf(3E) was almost diffusely distributed in the cytoplasm without K-Ras(G12V) coexpression, as shown above. Nevertheless, it was localized to the PM when K-Ras(G12V) was coexpressed (Fig 5B and C). Both DA-Raf(R52L) and DA-Raf(CRDm) were diffusely distributed in the cytoplasm at significant levels, irrespective of the coexpression of K-Ras(G12V) (Fig 5B and C). Accordingly, the DA-Raf cluster 2 bAAs do not directly participate in the binding of DA-Raf to active K-Ras on the PM, whereas the CRD is involved in the binding.

Moreover, BiFC analysis revealed that both DA-Raf(3E) and DA-Raf(CRDm) bound to K-Ras(G12V) at the PM less efficiently than did wt DA-Raf and that DA-Raf(R52L) hardly bound to K-Ras(G12V) (Fig 5D and E). Thus, although DA-Raf(3E) can also bind to active Ras as does wt DA-Raf in vitro, its binding efficiency is restricted in cells. FRAP analysis further showed that the recoveries of photobleached DA-Raf(3E) and DA-Raf(R52L) in the presence of K-Ras(G12V) at the PM were much faster than that of wt DA-Raf with K-Ras(G12V) (Fig 5F and G). Together, these results indicate that the PM localization of DA-Raf is cooperatively regulated by the interaction with PS in the PM and the binding to active Ras. In this context, the DA-Raf cluster 2 bAAs play an indirect role in the binding of DA-Raf to active Ras.

### RBD-mediated PM localization of DA-Raf interferes with Ras-dependent translocation of Raf to the PM

We next addressed whether the RBD-mediated PM localization of DA-Raf interferes with Ras-dependent translocation of Raf to the PM. As shown above, B-Raf coexpressed with K-Ras(G12V) was localized to the PM at a high level. However, further coexpression of DA-Raf together with K-Ras(G12V) greatly reduced the extent of PM localization of B-Raf (Fig 6A and C). On the other hand, coexpression of DA-Raf(3E) moderately reduced the PM localization level of B-Raf. When DA-Raf(3E) linked with the PM-anchoring myristoyl group (Myr) [Myr–mCherry–DA-Raf(3E)] was coexpressed, the PM localization level of B-Raf was greatly reduced as when DA-Raf was coexpressed (Fig 6A and C).

The unphosphorylatable CR2 mutant B-Raf(S348A) coexpressed with K-Ras(G12V) was also localized to the PM at a high level. Further coexpression of DA-Raf together with K-Ras(G12V) greatly abrogated the PM localization of B-Raf(S348A) (Fig 6B and C). However, coexpression of DA-Raf(3E) barely reduced the PM localization level of B-Raf(S348A). Nevertheless, coexpression of Myr–DA-Raf(3E) highly interfered with the PM localization of

---

**(C, D)** BiFC fluorescence intensity in the analysis of (C). The values are means ± SEM. **(E)** Localization of the unphosphorylatable CR2 mutants of Raf proteins in the absence of K-Ras(G12V). Shown are the distributions of the EGFP–A/B/C-Raf CR2 mutants in comparison with the PM in MDCK cells. Scale bar, 10 $\mu m$. **(F)** Localization of the unphosphorylatable CR2 mutants of Raf proteins in the presence of K-Ras(G12V). Shown are the distributions of the EGFP–A/B/C-Raf CR2 mutants and TagBFP–K-Ras(G12V) in comparison with the PM. **(E, F, G)** The PM localization ratio of the Raf protein CR2 mutants in the analyses of (E, F). The PM localization ratio is indicated as Fig 1E. **(H)** In vitro binding of Raf proteins, their CR2 mutants, and DA-Raf to K-Ras analyzed by a pull-down assay. The binding of EGFP-tagged A/B/C-Raf, their CR2 mutants, C-Raf(R89L), and DA-Raf in HeLa cell lysates to GST–K-Ras–GTPγS was detected by immunoblotting. **(I)** In vivo binding of the Raf protein CR2 mutants and DA-Raf to K-Ras(G12V) analyzed by BiFC analysis. The binding between KGN–K-Ras(G12V) and KGC-tagged A/B/C-Raf CR2 mutants or DA-Raf in MDCK cells was detected by BiFC. The PM staining is also shown. **(I, J)** BiFC fluorescence intensity in the analysis of (I). KGC–C-Raf(R89L) was used as a negative control. Source data are available for this figure.

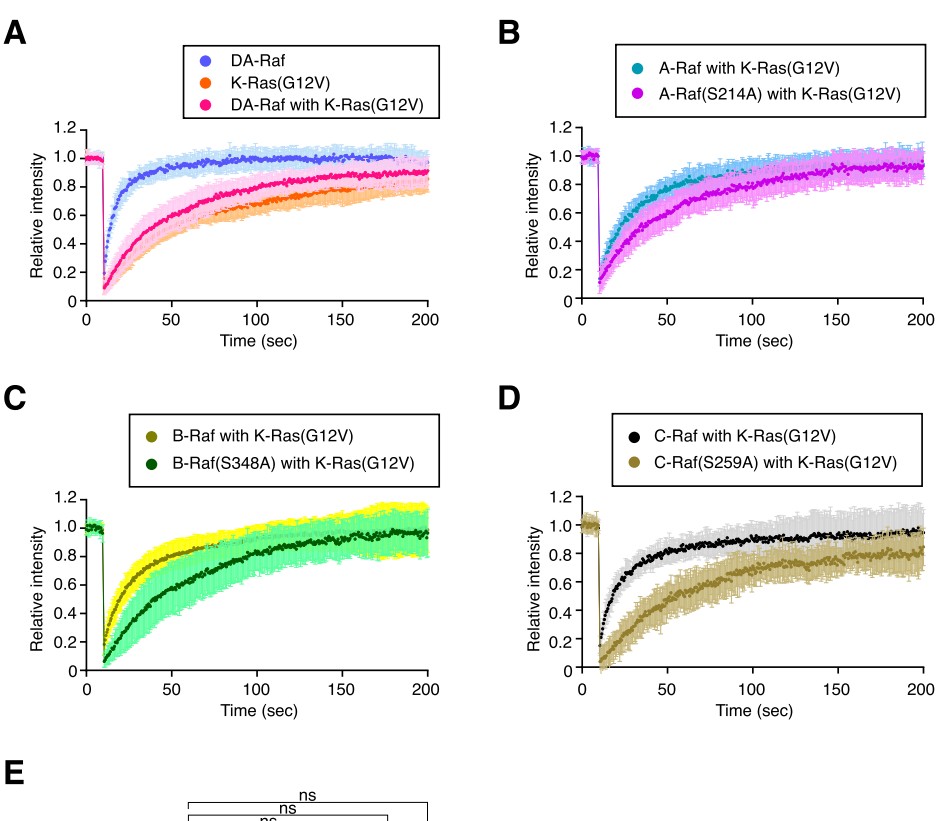

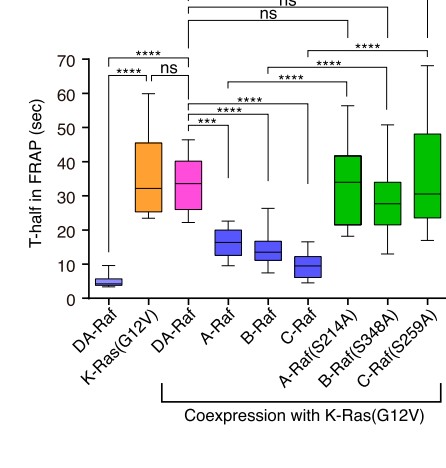

**Figure 3. Efficient binding of DA-Raf to active K-Ras at the PM because of the absence of CR2.**
**(A)** Fluorescence intensity in FRAP analyses of DA-Raf, K-Ras(G12V), and DA-Raf with K-Ras(G12V) on the PM. **(B)** Fluorescence intensity in FRAP analyses of A-Raf/A-Raf(S214A) with K-Ras(G12V) on the PM. **(C)** Fluorescence intensity in FRAP analyses of B-Raf/B-Raf(S348A) with K-Ras(G12V) on the PM. **(D)** Fluorescence intensity in FRAP analyses of C-Raf/C-Raf(S259A) with K-Ras(G12V) on the PM. Fluorescence intensity was measured at each time point on the PM of transfected MDCK cells (n = 15). The values are means ± SD. **(A, B, C, D, E)** Quantification of $T_{1/2}$ in the FRAP analyses of (A, B, C, D). The box limits indicate the 25th and 75th percentiles, the centerlines indicate means, and the whiskers represent the minimum and maximum values. Source data are available for this figure.

B-Raf(S348A), as did DA-Raf coexpression (Fig 6B and C). BiFC analysis further showed that B-Raf(S348A) bound to K-Ras(G12V) at the PM without the DA-Raf expression, but that B-Raf(S348A) hardly bound to K-Ras(G12V) under the DA-Raf expression (Fig 6D and E). B-Raf(S348A) bound to K-Ras(G12V) at the PM when DA-Raf(3E) was coexpressed. In contrast, it did not when Myr–DA-Raf(3E) was coexpressed (Fig 6D and E). Taken together, these results imply that DA-Raf predominates over B-Raf in the binding to active Ras at the PM, even if B-Raf adopts an unclosed conformation through unphosphorylated CR2. This is probably attributable to the efficient PM-associating properties of DA-Raf, which are mediated by its cluster 2 bAAs. The efficient PM-associating properties of DA-Raf may lead to the predominant Ras-binding ability of DA-Raf over B-Raf.

## Stable PM association of DA-Raf prevents Raf dimerization leading to the ERK pathway

Active Ras-induced Raf dimerization leads to the ERK pathway activation. Particularly, the B-Raf–C-Raf heterodimer exerts higher activity than their homodimers (Weber et al, 2001; Rushworth et al, 2006). Thus, we further explored whether the efficient PM association of DA-Raf interferes with the binding of B-Raf–C-Raf heterodimer to Ras and with the ERK pathway activation through Raf proteins. BiFC analysis showed that the heterodimer of B-Raf(S348A) and C-Raf was formed at the PM when K-Ras(G12V) was coexpressed. However, coexpression of DA-Raf together with K-Ras(G12V) abolished the B-Raf(S348A)–C-Raf heterodimer formation at the PM (Fig 7A and B). Although coexpression of

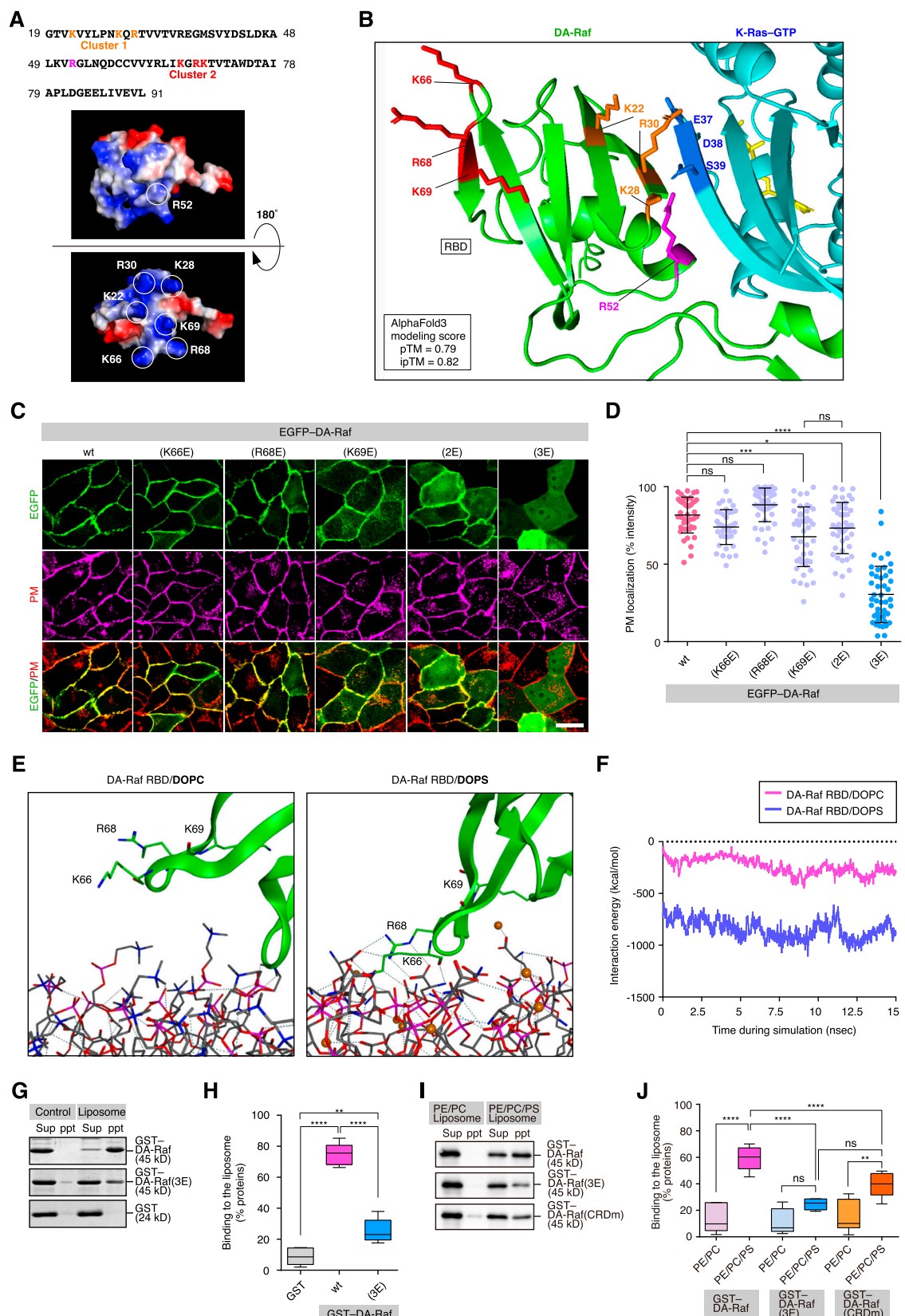

**Figure 4. Interaction of DA-Raf with PM phosphatidylserine via the basic amino acid cluster in the Ras-binding domain (RBD).**
**(A)** The amino acid sequence and 3D structure of DA-Raf RBD. Electrostatic potential surfaces of the 3D structures are shown in blue (basic region) and red (acidic region). Circles indicate the side chains of R52 in the central binding site to Ras (upper panel), K22, K28, R30 in cluster 1, and K66, R68, K69 in cluster 2 (lower panel).

DA-Raf(3E) barely interfered with the PM-localized heterodimer, coexpression of Myr–DA-Raf(3E) impeded the heterodimer formation at the PM, as did DA-Raf coexpression (Fig 7A and B).

We then analyzed the activation of ERK1/2 by fluorescence microscopy. Expression of K-Ras(G12V) highly induced activating phosphorylation of ERK1/2. However, coexpression of DA-Raf with K-Ras(G12V) blocked the ERK1/2 phosphorylation (Fig 7C and D). Although coexpression of DA-Raf(3E) barely inhibited the ERK1/2 phosphorylation, coexpression of Myr–DA-Raf(3E) impaired the ERK1/2 phosphorylation, as did DA-Raf coexpression (Fig 7C and D). These results indicate that the stable PM association of DA-Raf, which is brought about by the efficient PS association and cooperative or synergistic Ras-binding of DA-Raf, interferes with the B-Raf–C-Raf heterodimer formation by active Ras. Consequently, the stable PM association of DA-Raf prevents the ERK pathway activation. Fig 8A–D summarizes the results presented in this report.

## Discussion

The Ras-activated ERK pathway regulates a variety of cellular, physiological, and pathological events. The activity of the Ras–ERK pathway is strictly controlled by the coordinated action of both positive and negative regulators. Among the various negative regulators, DA-Raf uniquely acts as an intrinsic dominant-negative antagonist of the Ras–ERK pathway, thereby exerting its cellular and physiological functions (Yokoyama et al, 2007; Endo, 2020). However, the mechanisms of how DA-Raf fulfills the dominant-negative function on Raf proteins remain to be clarified. We have elucidated here this unsolved issue.

Ras activation and the subsequent PM localization of Raf proteins are a prerequisite for the activation of Raf proteins (Matallanas et al, 2011; Lavoie & Therrien, 2015; Terrell & Morrison, 2019). Indeed, Raf proteins were diffusely distributed in the cytoplasm in the absence of active Ras, whereas they were mobilized to the PM in the presence of PM-associated active K-Ras. On the other hand, DA-Raf was predominantly localized to the PM, regardless of the presence of active K-Ras. A substantial amount of DA-Raf remained at the PM even under the dominant-negative K-Ras expression. Moreover, DA-Raf bound to active K-Ras much more efficiently than did Raf proteins in cells. Therefore, although the PM localization of Raf proteins depends on Ras activity, DA-Raf has properties to essentially associate with the PM irrespective of

Ras activity. DA-Raf can bind to active Ras much more efficiently than do Raf proteins in cells, probably because of its efficient PM association properties.

The Raf protein CR2 contains autoinhibitory phosphorylation sites involved in the negative regulation of Ras-binding and Raf kinase activity, whereas DA-Raf lacks the CR2 and CR3. We have shown here that DA-Raf can bind to active K-Ras at the PM much more efficiently than Raf proteins, owing to the absence of the CR2. This notion is supported by our results that the unphosphorylatable CR2 mutants of Raf proteins stably bound to active K-Ras on the PM as did DA-Raf in cells. The efficient and predominant binding of DA-Raf to active K-Ras over Raf proteins leads to the stable PM association of DA-Raf.

Previous studies have shown that C-Raf interacts with membrane PS through a cluster of basic amino acids, R143, K144, K148, and K157, flanked by hydrophobic residues in the CRD (Improta-Brears et al, 1999; Travers et al, 2018; Fang et al, 2020; Tran et al, 2021). B-Raf CRD has a higher PS-binding ability than C-Raf CRD, because of certain amino acids specific to B-Raf CRD (Spencer-Smith et al, 2022). Although these amino acids are relatively well conserved in A-Raf, there have been no reports regarding the interaction of A-Raf CRD with membrane PS so far as we know. Thus, it is required to clarify whether and how A-Raf CRD interacts with membrane PS. We have elucidated here that K66 and R68 in the cluster 2 bAAs in DA-Raf RBD interact with PS in the PM by forming hydrogen bonds between them. This interaction is essential for the association of DA-Raf with the PM, regardless of Ras activity. The corresponding bAAs in A-Raf RBD have been shown to contact the membrane surface when A-Raf binds to active K-Ras mutants (Mazhab-Jafari et al, 2015). Because A-Raf RBD and DA-Raf RBD are identical, this report corroborates our findings. We have further shown here that R103, K104, and K117 in DA-Raf CRD also engage in the interaction with membrane PS. When Ras is not active on the PM, however, intact A-Raf, as well as B-Raf and C-Raf, cannot be associated with the PM because of its autoinhibitory closed conformation. Therefore, predominating over Raf proteins, DA-Raf is localized to the PM via the RBD cluster 2 bAAs and the CRD bAAs, regardless of Ras activity.

According to the MD simulation model, K66 and R68 in the cluster 2 bAAs interact with PS in the PM, whereas K69 appears to be involved in an intramolecular interaction within RBD rather than a direct interaction with PS. However, the PM localization of DA-Raf(K69E) is reduced to some degree, as is that of DA-Raf(2E), and that of DA-Raf(3E) is highly disrupted. Thus, the K69-mediated intramolecular interaction is likely to be critical to the PM

**(B)** 3D structure of the binding interface in the DA-Raf and K-Ras–GTP complex predicted with AlphaFold3. Shown are the main chains of DA-Raf (green) and K-Ras (cyan), DA-Raf R52 (magenta), cluster 1 bAAs (orange), cluster 2 bAAs (red), and GTP (yellow). Amino acid side chains are also shown by lines in the specified colors. **(C)** Localization of DA-Raf and its bAA cluster 2 mutants in MDCK cells. MDCK cells were transfected with EGFP–DA-Raf/DA-Raf cluster 2 mutants. Shown are the distributions of the EGFP-tagged proteins in comparison with the PM. Scale bar, 10 $\mu$m. **(C, D)** The PM localization ratio of DA-Raf and its bAA cluster 2 mutants in the analyses of (C). The PM localization ratio is indicated as Fig 1E. **(E)** Interaction of the cluster 2 bAAs in DA-Raf RBD with DOPC and DOPS shown by MD simulations. Dotted lines represent the interactions by hydrogen bonds. **(F)** The interaction energy between DA-Raf RBD and DOPC or DOPS bilayer shown by MD simulations. **(G)** In vitro interaction between DA-Raf/DA-Raf(3E) and PS-containing liposomes analyzed by a liposome-binding assay. Coprecipitation of GST–DA-Raf/DA-Raf(3E) with total bovine brain lipid liposomes was detected by SDS–PAGE. **(G, H)** Quantification of the binding between DA-Raf/DA-Raf(3E) and liposomes in the analysis of (G). The box, centerline, and whiskers represent the elements as described in Fig 3E legend. **(I)** In vitro interaction between DA-Raf/DA-Raf(3E)/(CRDm) and PE/PC or PE/PC/PS liposomes analyzed by a liposome-binding assay. Coprecipitation of GST–DA-Raf/DA-Raf(3E)/(CRDm) with PE/PC or PE/PC/PS liposomes was detected by immunoblotting with the anti-DA-Raf pAb. **(I, J)** Quantification of the binding between DA-Raf/DA-Raf(3E)/(CRDm) and the liposomes in the analysis of (I). The box, centerline, and whiskers represent the elements as described in Fig 3E legend.
Source data are available for this figure.

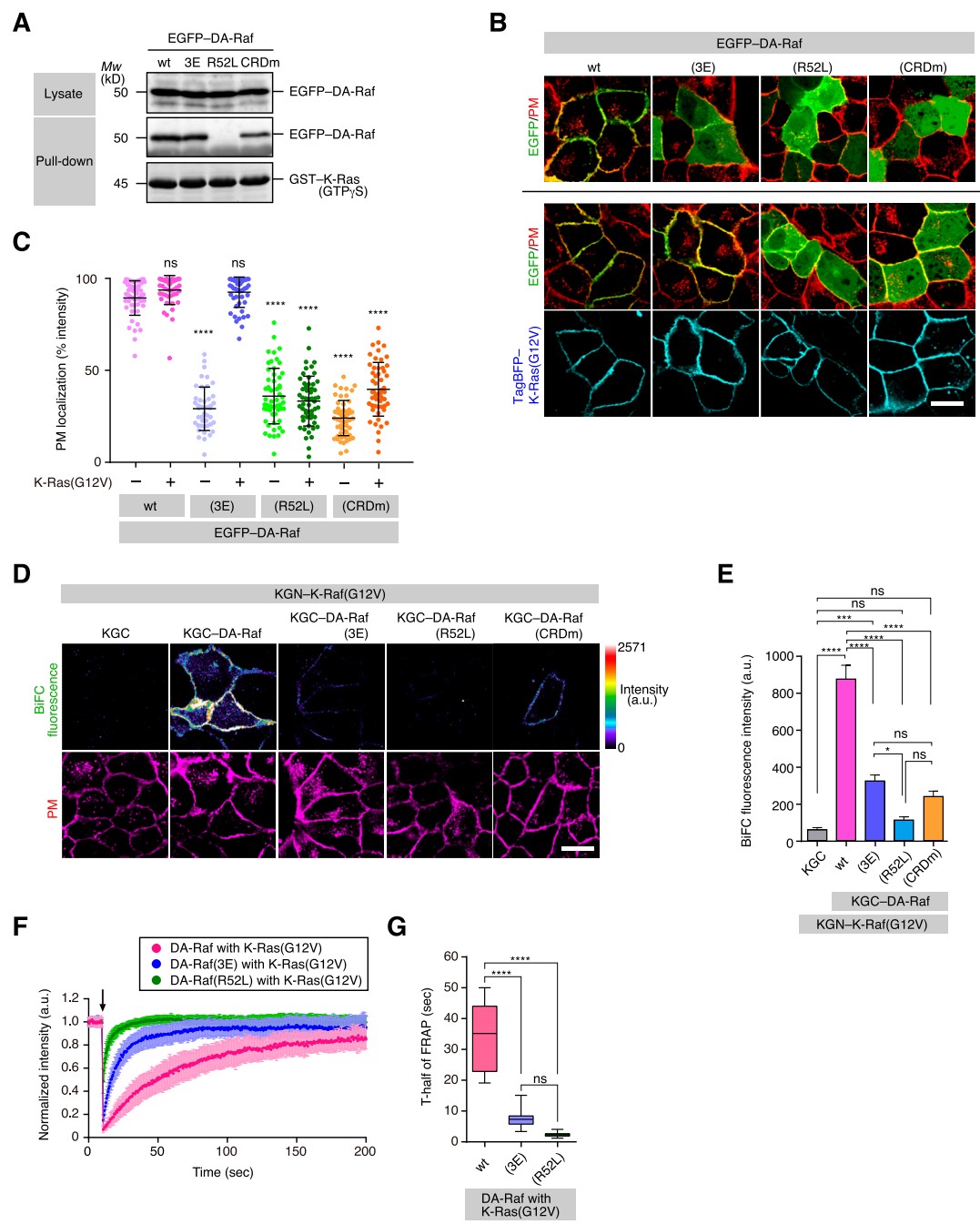

**Figure 5. PM localization of DA-Raf by the interaction with PS and the binding to active Ras.**
**(A)** In vitro binding of DA-Raf and its Ras-binding domain (RBD) mutants to K-Ras analyzed by a pull-down assay. The binding of EGFP–DA-Raf/DA-Raf(3E)/(R52L)/ (CRDm) in HeLa cell lysates to GST–K-Ras–GTPγS was detected by immunoblotting. **(B)** Localization of DA-Raf and its RBD mutants in MDCK cells expressing K-Ras(G12V). Shown are the distributions of EGFP–DA-Raf/DA-Raf(3E)/(R52L)/(CRDm) and TagBFP–K-Ras(G12V) in comparison with the PM. Scale bar, 10 μm. **(B, C)** The PM localization ratio of DA-Raf and its RBD mutants in the analyses of (B). The PM localization ratio is indicated as Fig 1E. **(D)** In vivo binding of DA-Raf and its RBD mutants to K-Ras(G12V) analyzed by BiFC analysis. The binding between KGN–K-Ras(G12V) and KGC–DA-Raf/DA-Raf(3E)/(R52L)/(CRDm) in MDCK cells was detected by BiFC. The PM staining is also shown. **(D, E)** BiFC fluorescence intensity in the analysis of (D). The values are means ± SEM. **(F)** Fluorescence intensity in FRAP analyses of DA-Raf, DA-Raf(3E), and DA-Raf(R52L) with K-Ras(G12V) on the PM. The values are means ± SD. **(F, G)** Quantification of $T_{1/2}$ in the FRAP analyses of (F). Source data are available for this figure.

localization of DA-Raf. On the other hand, the MD simulation model was constructed by applying a 100% PS membrane, whereas the inner leaflet of the PM is composed of ~20% PS. This difference

might account for the notion that K69 is involved in the intra-molecular interaction and not in direct interaction with PS. In addition, if another conformer in which K69 is released from the

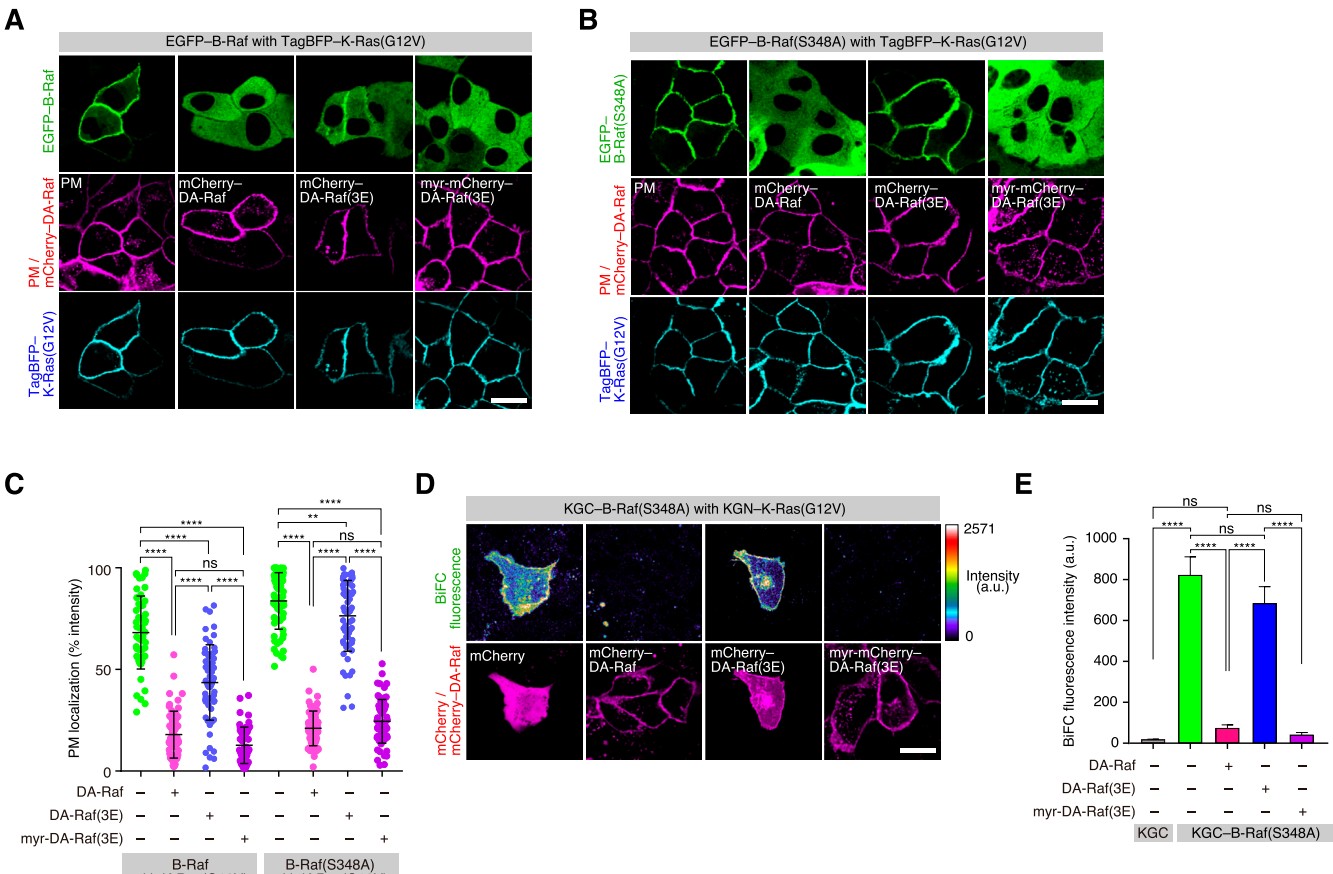

**Figure 6. Interference by Ras-binding domain-mediated PM localization of DA-Raf with Ras-dependent translocation of B-Raf to the PM.**
**(A)** Localization of B-Raf in MDCK cells expressing K-Ras(G12V) and DA-Raf/DA-Raf(3E)/Myr–DA-Raf(3E). Shown are the distributions of EGFP–B-Raf, mCherry–DA-Raf/ DA-Raf(3E) or Myr–mCherry–DA-Raf(3E), and TagBFP–K-Ras(G12V) in comparison with the PM. Scale bar, 10 μm. **(B)** Localization of B-Raf(S348A) in MDCK cells expressing K-Ras(G12V) and DA-Raf/DA-Raf(3E)/Myr–DA-Raf(3E). Shown are the distributions of EGFP–B-Raf(S348A), mCherry–DA-Raf/DA-Raf(3E) or Myr–mCherry–DA-Raf(3E), and TagBFP–K-Ras(G12V) in comparison with the PM. **(A, B, C)** The PM localization ratio of B-Raf and B-Raf(S348A) in the analyses of (A, B). The PM localization ratio is indicated as Fig 1E. **(D)** In vivo binding of B-Raf(S348A) to K-Ras(G12V) in the presence of DA-Raf/DA-Raf(3E)/Myr–DA-Raf(3E) analyzed by BiFC analysis. The binding between KGN–K-Ras(G12V) and KGC–B-Raf(S348A) in the presence of mCherry–DA-Raf/DA-Raf(3E) or Myr–mCherry–DA-Raf(3E) in MDCK cells was detected by BiFC. **(D, E)** BiFC fluorescence intensity in the analysis of (D). The values are means ± SEM.
Source data are available for this figure.

intramolecular interactions is selected in MD simulations, K69 might also directly interact with PS. Moreover, because active Ras should be present in the cells that we analyzed DA-Raf localization, MD simulations with active Ras are essential for the elucidation of the intracellular interaction of DA-Raf with PS in the PM via the RBD cluster 2 bAAs and the CRD bAAs.

Although the RBD cluster 2 bAAs and the CRD bAAs in DA-Raf are crucial for the PM association of DA-Raf, the binding to active K-Ras is also required for the PM localization of DA-Raf, as we have also shown here. This implies that the PM localization of DA-Raf is cooperatively regulated by the interaction with PS in the PM and the binding to active Ras. Nevertheless, the interaction of DA-Raf with PS may be indispensable for the binding to active Ras. In contrast, Ras activation triggers the binding of Raf proteins to Ras, leading to their conformational change for activation (Matallanas et al, 2011; Lavoie & Therrien, 2015; Terrell & Morrison, 2019; Spencer-Smith & Morrison, 2024). Then, Raf proteins interact with PS in the PM via the CRD, which is critical

for the direct association of Raf with the PM. Thus, the triggering mechanisms of PM localization are distinct between Raf proteins and DA-Raf.

We have further shown here that DA-Raf prevents the binding of B-Raf to active K-Ras and B-Raf–C-Raf dimer formation at the PM, even if B-Raf adopts unclosed conformations because of unphosphorylated CR2. This is ascribable to the efficient PM-associating properties of DA-Raf, which are mediated by the RBD cluster 2 bAAs and the CRD bAAs. The efficient PM-associating properties of DA-Raf may lead to the predominant Ras-binding ability of DA-Raf over Raf proteins. Once DA-Raf binds to active Ras, Raf proteins can no longer bind to the Ras. In addition, DA-Raf lacks the CR3 that represents the kinase domain. Consequently, DA-Raf can exert its dominant-negative function on Raf proteins and thereby interferes with the Ras–ERK pathway (Fig 8). To comprehensively reveal the dominant-negative function of DA-Raf in vivo, important things are elucidation of the kinetic stability of DA-Raf at the membrane and positive or negative feedback regulation of

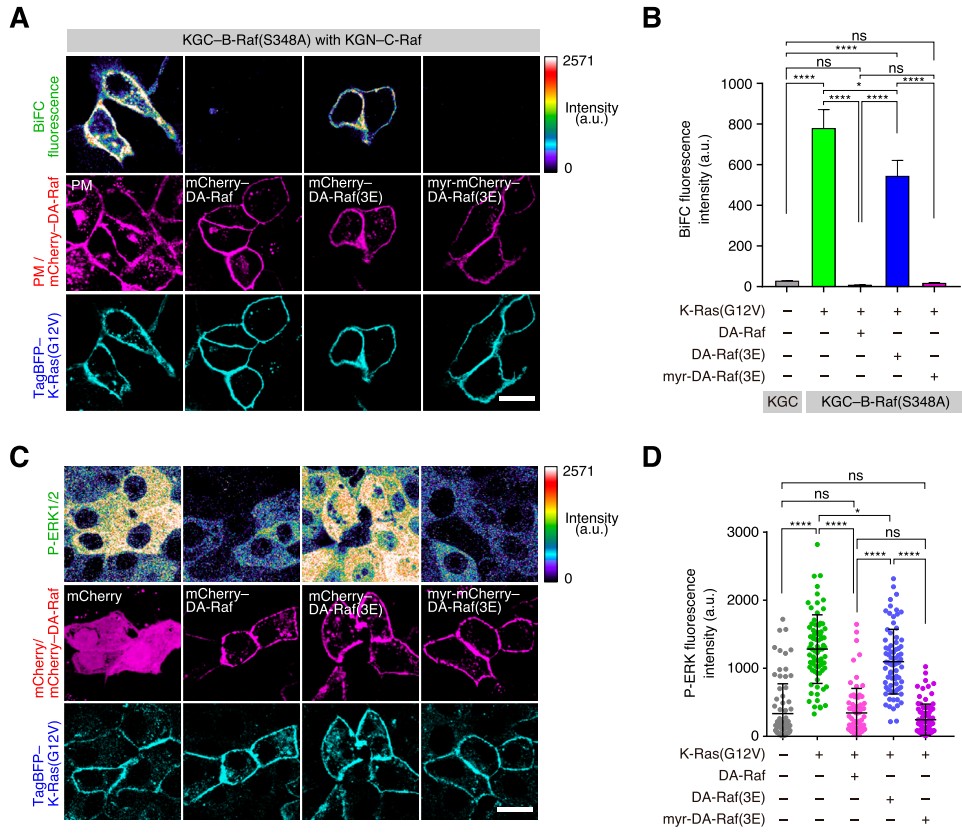

**Figure 7. Prevention of Raf dimerization and ERK activation by stable PM association of DA-Raf.**
**(A)** In vivo binding between B-Raf(S348A) and C-Raf in the presence of K-Ras(G12V) and DA-Raf/DA-Raf(3E)/Myr–DA-Raf(3E) analyzed by BiFC analysis. The binding between KGN–C-Raf and KGC–B-Raf(S348A) in the presence of TagBFP–K-Ras(G12V) and mCherry–DA-Raf/DA-Raf(3E) or Myr–mCherry–DA-Raf(3E) in MDCK cells was detected by BiFC. Scale bar, 10 $\mu$m. **(B, D)** BiFC fluorescence intensity in the analysis of (D). The values are means ± SEM. **(C)** K-Ras(G12V)-induced ERK activation in the presence of DA-Raf/DA-Raf(3E)/Myr–DA-Raf(3E). MDCK cells were cotransfected with TagBFP–K-Ras(G12V) and mCherry–DA-Raf/DA-Raf(3E) or Myr–mCherry–DA-Raf(3E), and phospho-ERK1/2 (P-ERK1/2) was detected by the staining with the anti-phospho-ERK1/2 mAb. Shown are the P-ERK1/2 level (in rainbow colors), the localization of mCherry–DA-Raf/DA-Raf(3E) or Myr–mCherry–DA-Raf(3E), and TagBFP–K-Ras(G12V). **(D)** P-ERK1/2 fluorescence intensity in the analysis of (D). The horizontal lines indicate means ± SD.
Source data are available for this figure.

DA-Raf, if exists, in relation to Raf protein regulation. We need to examine these matters in our next studies.

Extracellular signal-activated Ras acts on various effector proteins, including PI3K, RalGEFs, and Raf proteins, to conduct a variety of cellular and physiological functions (Karnoub & Weinberg, 2008; Cox & Der, 2010). It is crucial to determine whether the binding of DA-Raf to active Ras blocks the binding of these effector proteins to active Ras. If DA-Raf prevents these effector proteins from binding to active Ras, DA-Raf should hinder the signaling pathways induced by these effector proteins. Exogenously overexpressed DA-Raf suppresses ERK activity but not Akt activity in oncogenic v-*Kras*-transformed fibroblasts and in skeletal muscle myoblasts (Yokoyama et al, 2007). Rather, prominent elevation of DA-Raf expression and the subsequent Akt activation occur during skeletal myocyte differentiation (Takahashi et al, 2019). These findings suggest that DA-Raf does not affect the binding of PI3K to Ras, which induces Akt activation. One reason for this is ascribed to that the sites in Ras involved in the binding of Raf and PI3K p110 catalytic subunit are partially overlapping but distinct (Pacold et al, 2000; Vetter & Wittinghofer, 2001; Mozzarelli et al, 2024). Thus, DA-Raf might interfere with the binding of Raf proteins but not with that of PI3K p110. Another reason for the inability of DA-Raf to inactivate Akt is attributed to that PI3K can be activated by Ras-independent mechanisms, such as trimeric G-protein G$\beta\gamma$ subunit-mediated activation (Engelman et al, 2006). On the other hand, because the RalGEF-binding site overlaps with the Raf-binding site (Vetter et al, 1999; Vetter & Wittinghofer, 2001; Mozzarelli et al, 2024), DA-Raf might

prevent the binding of RalGEFs to Ras and Ras-induced RalGEF signaling. Scrutinization of the Ras-binding properties of Ras effectors in cells and in vivo in the presence of DA-Raf may reveal novel cellular and physiological functions of DA-Raf.

DA-Raf exerts tumor-suppressing and invasion-suppressing functions to cancer cells with oncogenic *KRAS* mutations (Yokoyama et al, 2007; Kanno et al, 2018; Matsuda et al, 2024), DA-Raf also induces skeletal myocyte differentiation (Yokoyama et al, 2007; Takahashi et al, 2019) and counteracts skeletal muscle atrophy and sarcopenia caused by TGF-$\beta$ superfamily protein-induced non-Smad Ras–ERK pathway (Masuzawa et al, 2022). DA-Raf further participates in lung alveolar septum formation through myofibroblast differentiation (Watanabe-Takano et al, 2014) and EMT from alveolar epithelial type 2 cells to myofibroblasts (Watanabe-Takano et al, 2015). Therefore, this study may provide helpful information to develop therapies for cancers, muscle atrophy and sarcopenia, and lung diseases such as chronic obstructive pulmonary disease (COPD).

## Materials and Methods

### Plasmid construction and introduction of mutations

cDNAs encoding mouse K-Ras, DA-Raf, A-Raf, B-Raf, human C-Raf, and mouse 14-3-3$\beta$ were cloned by PCR and inserted into pEGFP,

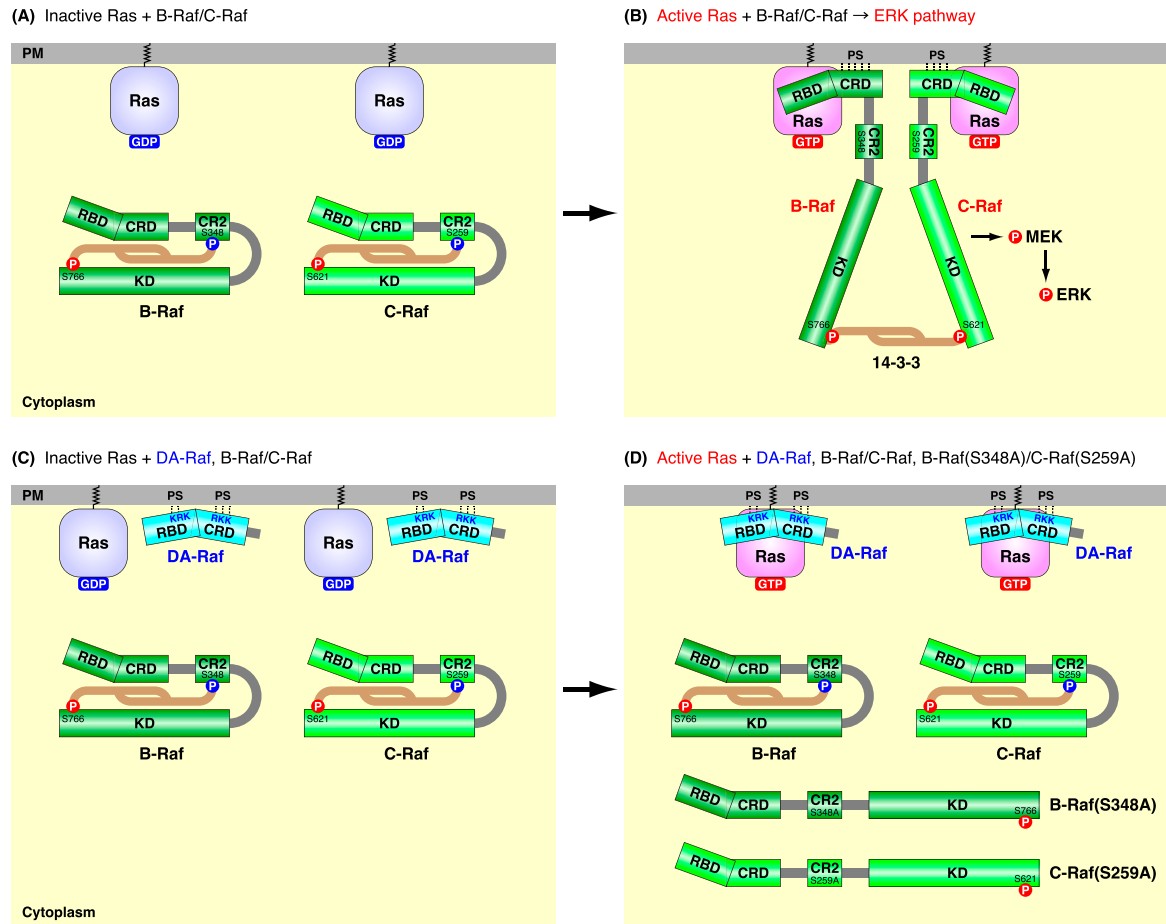

**Figure 8. Mechanisms of the dominant-negative function of DA-Raf to Raf proteins to prevent the Ras–ERK pathway.**
**(A)** When Ras is inactive and DA-Raf is absent, B-Raf and C-Raf fold into autoinhibitory closed conformations via 14-3-3 dimer binding. They are located in the cytoplasm. **(B)** When Ras is active and DA-Raf is absent, B-Raf and C-Raf bind to Ras via their Ras-binding domains (RBDs) and Cys-rich domains (CRDs). The Ras-binding elicits CR2 dephosphorylation and 14-3-3 release from the CR2, thereby removing B/C-Raf autoinhibition. The B/C-Raf CRD interacts with PS in the PM. Subsequently, B/C-Raf is activated by dimerization through 14-3-3 dimer binding between their C-terminal phospho-Ser. The activated B/C-Raf dimer induces the ERK pathway. **(A, C)** When Ras is inactive and DA-Raf is present, B-Raf and C-Raf are inactive as in (A). DA-Raf is substantially localized to the PM through the interaction of the RBD cluster 2 bAAs and the CRD bAAs with PS. **(D)** When Ras is active and DA-Raf is present, DA-Raf associated with the PM efficiently binds to Ras via the RBD and CRD. Thus, DA-Raf dominant negatively interferes with the binding of Raf proteins. **(A)** B-Raf and C-Raf are inactive as in (A). Even if they adopt unclosed conformations via unphosphorylated CR2, as B-Raf(S348A) and C-Raf(S259A), DA-Raf predominates over them in the binding to active Ras. Consequently, DA-Raf prevents the Ras–ERK pathway.

pmCherry (Clontech), pTagBFP (Evrogen), phmKGN, phmKGC vectors (ColalHue Fluo-chase kit, MBL). Synthetic oligonucleotides encoding the myristoylation site of mouse c-Src (MGSSKSKPKDP) were cloned into the NheI sites of pEGFP and pmCherry vectors to fuse with the N-terminus of fluorescent proteins.

Mutations in cDNAs were introduced with mutagenic oligonucleotide primers by using PrimeSTAR Mutagenesis Basal Kit (Takara). These mutants were K-Ras(G12V) and K-Ras(S17N); C-Raf(R89L) and CR2 mutants of A-Raf(S214A), B-Raf(S348A) [corresponding to human B-Raf(S365A)], C-Raf(S259A), C-Raf(S259A/S621A); DA-Raf(R52L), DA-Raf cluster 2 bAA mutants of (K66E), (R68E), (K69E), (K66E/R68E) = (2E), (K66E/R68E/K69E) = (3E), and DA-Raf CRD mutant (CRDm) of (R103E/K104E/K117E). Sequences of the myristoylation site oligonucleotides and the mutagenic oligonucleotide primers used are listed in Table S1.

## Cell culture and transfection

MDCK epithelial cells (JCRB9029) and human cervical adenocarcinoma HeLa cells (JCRB9004) were obtained from JCRB Cell Bank. They were cultured in high-glucose DMEM (D7777; Sigma-Aldrich) containing 10% FBS (10270106; Thermo Fisher Scientific). MDCK cells were transfected by electroporation with Electroporator NEPA-21 (Nepa Gene). Poring pulse parameters of the electroporation optimized for MDCK cells were set as follows: voltage, 150 V; pulse length, 5 msec; pulse interval, 50 msec; pulse number, 2; decay rate, 10%; polarity switching, plus. For electroporation, $5 \times 10^6$ cells and 20 $\mu$g of DNA per cuvette were used. After the electroporation, the cells were plated onto 35-mm glass-based dishes (3910-035; Iwaki). HeLa cells on 35-mm glass-based dishes ($6 \times 10^5$ cells/dish) were transfected by using Lipofectamine 3000 (Thermo Fisher Scientific). Almost equimolar amounts of plasmid constructs of Raf

proteins and DA-Raf were used for transfection to adjust their expression levels. The medium was replaced with fresh DMEM containing 10% FBS 4 h after the transfection.

### GST-tagged protein expression and purification

GST-tagged proteins were expressed in insect Sf21 cells (Thermo Fisher Scientific) with the Bac-to-Bac Baculovirus Expression System (Thermo Fisher Scientific). The *E. coli* strain DH10Bac was transformed with pFastBac1/GST-proteins. Sf21 cells in SF900-II medium on a 35-mm dish ($8 \times 10^5$ cells/dish) were transfected with the minipreparation of recombinant bacmid DNA by using Cellfectin II Reagent (Thermo Fisher Scientific). After maintaining the cells for 3–5 h at 28°C, the medium was replaced with a SF900-II medium containing 5% FBS, and the cells were further maintained for 72 h at 28°C. Baculovirus in the medium supernatant was amplified through three rounds of amplification according to Bac-to-Bac Baculovirus Expression System user guide (Thermo Fisher Scientific). Sf21 cells ($5 \times 10^7$ cells in 50 ml) were infected with recombinant baculovirus (1: 100 diluted) and cultured for 72 h at 28°C with shaking. The cells were collected by centrifugation at 130$g$ for 5 min at 4°C and lysed by sonication in 10 ml of Triton lysis buffer (0.1% Triton X-100 [not included for liposome-binding assay], 50 mM Tris–HCl [pH7.5], 150 mM NaCl, 5 mM EDTA, 0.1 mM PMSF, 10 μg/ml leupeptin, and 1 μg/ ml pepstatin A) on ice. The lysates were clarified by centrifugation at 22,140$g$ for 15 min at 4°C and mixed with 100 μl of Glutathione Sepharose 4B (GE Healthcare) for 2 h at 4°C with shaking. The mixture was washed five times with 10 ml of the lysis buffer by centrifugation at 130$g$ for 3 min at 4°C.

### Pull-down assay

GST-tagged protein-coupled Glutathione Sepharose 4B was suspended in 1 ml of the lysis buffer, packed in a 1.5-ml microcentrifuge tube, and subjected to GST pull-down assays as described previously (Takano et al, 2010; Kanno et al, 2018). For GST–K-Ras, GTPγS was added at a final concentration of 1 mM and incubated for 10 min at 37°C. HeLa cells transfected with EGFP–A-Raf/B-Raf/C-Raf or EGFP–DA-Raf/DA-Raf(3E)/DA-Raf(R52L) on a 60-mm dish were washed with PBS and then lysed with 500 μl of NP-40 lysis buffer (1% Nonidet P-40, 50 mM Tris–HCl [pH7.5], 100 mM NaCl, 15 mM $MgCl_2$, 5% glycerol, 1 mM DTT, 0.1 mM PMSF, 10 μg/ml leupeptin, 1 μg/ml pepstatin A, 2 mM $Na_3VO_4$, and 10 mM NaF) by pipetting on ice. The lysates were clarified by centrifugation at 22,140$g$ for 15 min at 4°C. They were mixed with GST–K-Ras–GTPγS-coupled Glutathione Sepharose 4B in 1 ml of NP-40 lysis buffer for 60 min at 4°C with mild shaking. The mixture was washed three times with 500 μl of NP-40 lysis buffer by centrifugation at 130$g$ for 1 min at 4°C. Forty μl of 2 × Laemmli's SDS sample buffer was added to the precipitated Glutathione Sepharose 4B beads and boiled for 3 min. The supernatants were subjected to SDS–PAGE, electrophoretically transferred to Immobilon-P PVDF membrane (IPVH00005; Merck Millipore), and analyzed by immunoblotting with anti-GFP pAb.

### Immunoblotting

The transferred PVDF membrane was treated with blocking buffer (TBS [20 mM Tris–HCl, pH7.5, 150 mM NaCl], 5% skimmed milk, 0.2% Tween 20, and 0.02% $NaN_3$) for 1 h at RT and rinsed twice with TBST (TBS and 0.2% Tween 20). The membrane was reacted with the primary antibody anti-GFP rabbit pAb (598; MBL) (1:1,000 diluted with Blocking buffer) for 1 h at RT, washed five times with TBST, reacted with the secondary antibody HRP–anti-rabbit goat IgG (7074; Cell Signaling Technology) (1:2,000 diluted with Blocking buffer) for 1 h at RT, and washed five times with TBST. The membrane was treated with Western Lighting Plus ECL Reagent (PerkinElmer) for 1 min and dried. Blotting bands were detected with ChemDoc XRS Plus System (Bio-Rad) with Image Lab Software Ver. 4.1 (Bio-Rad).

### Confocal fluorescence microscopy

MDCK and HeLa cells were transfected with pEGFP, pmCherry, pTagBFP, phmKGN, and phmKGC recombinant plasmids, as described above. Thirty-six to 48 h after the transfection, they were treated with 5 μg/ml CellMask Orange Plasma Membrane Stain (C10045; Thermo Fisher Scientific) in DMEM for 5 min to detect the PM. The fluorescence of these living cells on 35-mm glass-based dishes at 37°C were observed by using a confocal laser-scanning microscope FV1200 (Olympus) equipped with a PlanApo N 60× oil-immersion objective lens (NA 1.40) and a stage-top $CO_2$ incubator (INUG2-ONICS, Tokai Hit). Fluorescence images (512 × 512 pixels, 2 × zoom) were acquired with FV10-ASW viewer software Ver. 4.2b (Olympus).

To detect the phospho-ERK1/2 level, MDCK cells were cotransfected with pmCherry/DAraf and pTagBFP/K-Ras(G12V) and fixed with 4% PFA in PBS 48 h after the transfection. They were incubated with anti-phospho-ERK1/2 rabbit mAb (4370; Cell Signaling Technology) labeled with Zenon rabbit IgG labeling kit (Z25308; Thermo Fisher Scientific), and washed five times with PBS. The specimens were observed with the confocal laser-scanning microscope. The fluorescence intensity was measured and processed to rainbow colors (range 0–2,571) with MetaMorph software Ver. 7.8 (Molecular Devices).

The background intensity of each image was subtracted by using the background subtraction function of MetaMorph software, based on the fluorescence intensity of the cell region without fluorescent protein expression or unstained region. A threshold was determined on the basis of the PM staining intensity. Each cell expressing fluorochrome-tagged proteins was segmented by manual region selection. After the calibration and segmentation, fluorescence intensity on the PM and the whole cell was analyzed with the colocalization function of MetaMorph software. This was calculated by defining PM localization (%) as the fluorescence intensity of the EGFP-tagged protein that coexists with the PM staining among the fluorescence intensity of all the EGFP-tagged proteins in the cell. GraphPad Prism 7.0b software was used to plot the PM localization data.

## BiFC analysis

MDCK cells were cotransfected with the recombinant plasmids to express KGN–K-Ras(G12V) or KGN–14-3-3$\beta$ and KGC–DA/A/B/C-Raf or their mutants by electroporation. Living cells were observed by confocal fluorescence microscopy 48 h after the transfection, as described above. Fluorescence images were acquired with the same detection condition among experiments (lens, PlanApo N 60× oil-immersion objective lens [NA 1.40]; laser transmittance, 1%; HV, 648 V; gain, 3×; offset, 24%; size, 512 × 512 pixels; Kalman, 2 × line). Cells were incubated at 37°C in 5% $CO_2$.

## FRAP

MDCK cells were transfected with the recombinant plasmids to express EGFP–DA-Raf, EGFP–K-Ras(G12V), EGFP–A/B/C-Raf or their mutants, and TagBFP–K-Ras(G12V) by electroporation. Image data acquisition and laser regulation for FRAP were conducted 48 h after the transfection by using the confocal laser-scanning microscope FV1200 (Olympus) equipped with a PlanApo N 60× oil-immersion objective lens (NA 1.40) and a stage-top $CO_2$ incubator (INUG2-ONICS; Tokai Hit). Cells were incubated at 37°C in 5% $CO_2$. Fluorescence images were acquired with the FRAP application of FV10-ASW viewer software Ver. 4.2b (Olympus). Parameters of image acquisition were: 20 frames acquisition before photobleaching (a total of 500 frames); image size, 320 × 320 pixels (4 × zoom); frame rate, 0.5 sec/frame; bleaching radius, 10 pixels. After image acquisition, the fluorescent intensities of the bleached area, the total area of a cell expressing fluorescent proteins, and background area were measured with the region measurement function of MetaMorph software. These data were imported to easyFRAP software (Rapsomaniki et al, 2012) to estimate the $T_{1/2}$ from the fitting of each normalized curve with a double exponential equation. Each FRAP data were derived from at least individual 15-cell images.

## Liposome-binding assay

Liposome-binding assay was performed as described previously (Tsujita et al, 2006; Takano et al, 2008). Total bovine brain lipids (Folch fraction I, Avanti Polar Lipids) or PE, PC, and PS (Sigma-Aldrich) were used to prepare liposomes. The ratio of each phospholipid was PE:PC = 5:5 in PE/PC liposomes and PE:PC:PS = 4:4:2 in PE/PC/PS liposomes. They were dried under vacuum centrifugation, resuspended in XB (10 mM HEPES-NaOH [pH 7.9], 100 mM KCl, 2 mM $MgCl_2$, 0.2 mM $CaCl_2$, and 5 mM EGTA) with a vortex mixer, and hydrated for 1 h at 37°C. GST–DA-Raf/DA-Raf(3E) were expressed in Sf21 cells and purified with glutathione Sepharose 4B as described above. The purified GST-tagged proteins (2 $\mu$M each) were incubated with 10 $\mu$g of liposomes for 20 min at RT and centrifuged at 25,000$g$ for 30 min at 4°C with CS100GXL ultracentrifuge (Hitachi). The supernatants and precipitates were subjected to SDS–PAGE, and proteins were detected by Coomassie brilliant blue (CBB) R-250 staining or immunoblotting with the anti-DA-Raf pAb. The intensity of bands was densitometrically analyzed with ImageJ software (NIH) by selecting each band with the region tool after background intensity was manually subtracted.

## MD simulations and 3D structure prediction

Construction of DA-Raf RBD–lipid bilayer complexes and MD simulations were performed basically as described (Dickson et al, 2014). The solution NMR structure of human A-Raf/DA-Raf RBD (Zhao et al, 2005) was obtained from the RCSB Protein Data Bank (PDB) (https://www.rcsb.org). The 3D structure of DA-Raf RBD was illustrated by using PyMOL software (https://pymol.org) on the basis of the NMR structure. The complex of DA-Raf RBD and a phospholipid bilayer composed of 100% DOPC or DOPS was constructed using the CHARMM Membrane Builder GUI (Lee et al, 2016) and converted to Lipid11 PDB format by using the charmmlipid2amber.x script. Constant pressure and constant temperature (NPT) runs were performed on DA-Raf RBD–lipid bilayer complexes using the AMBER 14 package. Bonds involving hydrogen were constrained using the SHAKE algorithm, allowing a 2 fs time step. Structural data were recorded every 10 ps. PME was used to treat all electrostatic interactions with a real-space cutoff of 10 Å. A long-range analytical dispersion correction was applied to the energy and pressure. The non-bonded interaction energy in terms of electrostatic and Van der Waal's between DA-Raf RBD and lipid bilayer was calculated by NAMDEnergy plug-in in visual molecular dynamics (VMD) (Humphrey et al, 1996). All figures from the MD simulations of these complexes were produced by using MOE (version 2013, Chemical Computing Group). 3D structure prediction of DA-Raf and K-Ras–GTP complex was performed with AlphaFold 3 (Abramson et al, 2024; https://alphafoldserver.com). Amino acid sequences used in AlphaFold3 were obtained from UniPlot (https://www.uniprot.org). After performing the 3D structure prediction multiple times, the representative 3D structure, for which a predicted template modeling (pTM) score was 0.5 or higher and an interface pTM (ipTM) score was 0.8 or higher, was displayed by using PyMOL.

## Statistical analysis

The data of PM localization and BiFC analysis were acquired from ≥40 individual cells in ≥3 independent experiments. The FRAP data were obtained from 15 cells in each experiment. Protein–lipid binding was analyzed six times for each experiment. These data were analyzed with GraphPad Prism 7.0b software. Statistical analysis was conducted by one-way ANOVA followed by the Kruskal-Wallis test. $P$-values on the graphs are *$P < 0.05$; **$P < 0.01$; ***$P < 0.001$; ****$P < 0.0001$.

## Online supplemental material

Fig S1 shows the localization of DA-Raf and the Raf proteins in HeLa cells with or without the expression of K-Ras(G12V). Fig S2 shows structural models of DOPC/DOPS bilayers and DA-Raf RBD. Table S1 lists the sequences of the mutagenic oligonucleotide primers used in this study. Video 1 shows the MD simulations of interaction between DA-Raf RBD and DOPC bilayer. Video 2 shows the MD simulations of interaction between DA-Raf RBD and DOPS bilayer.

## Data Availability

The data underlying this study are openly available in Source Data and also from the corresponding author upon reasonable request.

### English proofreading

AI-based English proofreading of the text was conducted by using Grammarly EDU for Chiba University.

## Supplementary Information

## Acknowledgements

This work was supported by the Japan Society for the Promotion of Science (JSPS) KAKENHI (JP15H04348 to T Endo; JP17K07328 and JP21K06078 to K Takano); Kishimoto Foundation, Uehara Memorial Foundation, Naito Foundation, Mochida Memorial Foundation, and Takeda Science Foundation (to K Takano); Takeda Science Foundation, Uehara Memorial Foundation, and Mitsubishi Foundation, (to T Endo); Kobe University Collaborative Research (Biosignal Research Center 282007 and 201005 to K Takano).

### Author Contributions

K Takano: conceptualization, resources, data curation, formal analysis, supervision, funding acquisition, validation, investigation, visualization, methodology, project administration, and writing—original draft, review, and editing.
K Tsujita: investigation and writing—review and editing.
A Suganami: formal analysis.
T Nakamura: resources.
E Kanno: resources.
Y Tamura: formal analysis.
T Itoh: resources, methodology, and writing—original draft.
T Endo: conceptualization, funding acquisition, visualization, and writing—original draft, review, and editing.

### Conflict of Interest Statement

The authors declare that they have no conflict of interest.

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
