## [Reviewer comments · Life Science Alliance]

DA-Raf dominant-negatively regulates Raf by preferentially binding to the plasma membrane and Ras

Kazunori Takano, Kazuya Tsujita, Akiko Suganami, Takuhiko Nakamura, Emiri Kanno, Yutaka Tamura, Toshiki Itoh, and Takeshi Endo

DOI: <https://doi.org/10.26508/lsa.202503300>

Corresponding author(s): Kazunori Takano, Chiba University

Review Timeline:

Submission Date:	2025-03-10
Editorial Decision:	2025-04-02
Revision Received:	2025-08-05
Editorial Decision:	2025-09-08
Revision Received:	2025-09-17
Editorial Decision:	2025-09-19
Revision Received:	2025-09-25
Accepted:	2025-09-26

Scientific Editor: Sarita Hebbar

Transaction Report:

April 1, 2025

Re: Life Science Alliance manuscript #LSA-2025-03300-T

Dr. Kazunori Takano
Chiba University
Department of Biology, Graduate School of Science
1-33, Yayoi-cho, Inage-ku
Chiba, Chiba 263-8522
Japan

Dear Dr. Takano,

Thank you for submitting your manuscript entitled "DA-Raf dominant-negatively antagonizes Raf by its predominant binding to the plasma membrane and Ras" to Life Science Alliance. The manuscript was assessed by three expert reviewers, whose comments are appended to this letter.

Overall, the reviewers concluded that findings in this manuscript are of value to the community. However, they raised important concerns that need to be addressed for publication here. Some of these are listed here:

- All reviewers sought improved evidence for DA-Raf1 binding to PS, either through controlled liposome binding assays or through improved MD simulations. This concerns should be addressed in a manner of your choosing.
- Reviewers 1 and 3 remarked that pulldown experiments are lacking needed controls, and Reviewer 1 suggested specific mutant constructs.

Given the overall recommendations, we would like to invite you to submit a revised manuscript addressing the Reviewers' comments. Please include a letter addressing all the reviewers' comments point by point.

Thank you for this interesting contribution to Life Science Alliance. We are looking forward to receiving your revised manuscript.

Sincerely,

Sarita Hebbar, PhD
Scientific Editor
Life Science Alliance
<http://www.lsjournal.org>

B. MANUSCRIPT ORGANIZATION AND FORMATTING:

Reviewer #1 (Comments to the Authors (Required)):

This manuscript investigates the mechanisms by which DA-Raf1 exerts its dominant-negative function on the RAS-ERK signaling pathway. The authors provide evidence that DA-Raf1, a splicing isoform of ARAF, preferentially localizes to the plasma membrane due to the absence of the CR2 14-3-3 docking site and CR3 kinase domain which play key roles in RAF autoinhibition. They also propose that the plasma membrane localization of DA-Raf1 is at least in part facilitated by an interaction between a basic amino acid cluster in DA-Raf1's RAS-binding domain and phosphatidylserine in the plasma membrane. Finally, they show that the constitutive plasma membrane localization of DA-Raf allows it to more rapidly bind active RAS than full-length RAF proteins which are localized to the cytosol in their pre-signaling state. Thus, these properties allow DA-Raf1 to efficiently inhibit RAS signaling. The Endo lab have published extensively on the roles of DA-RAF1 in human development and disease and this study provides some valuable mechanistic insights into how DA-Raf1 exerts its dominant-negative function. In addition, the identification of a RBD membrane binding interface will be of broad interest to the RAS community. However, the authors should first address a number of critical issues, especially regarding the use of appropriate controls in biochemical assays:

Major comments:

1. Figures 1A shows that KRAS pulls down similar levels of DA-Raf1 as A/B/CRAF, however there are no controls to show that these are specific interactions. The same is true for figures 2B & 2H. Examples of negative controls for Figs 1A and 2H are RAS binding impaired mutations (CRAF R89L, BRAF R188L, ARAF/A-Raf1 R53L). Mutating both 14-3-3 sites to alanine, thus alleviating 14-3-3 binding to RAF, would be an appropriate control for Fig. 2B (e.g. CRAF S259A/S621A).
2. Figure 1D shows that expression of dominant negative KRAS 17N partially mislocalizes DA-Raf1 to the cytosol, yet in the absence of any KRAS co-expression (Fig 2A) DA-Raf1 is plasma membrane localized. To what mechanism do the authors attribute this effect?
3. Again, no negative controls for RAS binding to RAF are used in BiFC experiments 1G & 2I, or for 14-3-3 binding to RAF in Fig 2C. Given that BiFC creates essentially irreversible fusion of the split fluorophore such controls are especially important to show that these interactions are specific and not artifacts of the experimental approach.
4. The conclusions section contains unnecessary speculation. This section should be made concise and focus on the facts, e.g. "We have further shown here that DA-Raf prevents the binding of B-Raf and B-Raf-C-Raf dimer (probably that of A-Raf as well)" This is conjecture as ARAF dimerization was not tested. Especially given that previous studies indicate very low levels of A-RAF dimerization (Mol Cell 2013 Feb 21;49(4):751-8.).
5. On page 8 the authors state: "the interaction of the cluster 2 BAAs with PS is responsible for the PM localization of DA-Raf independently of Ras activation" however, the ability of these amino acids to interact with PS is not directly tested (presumably Folch fraction I used for liposomes contains other phospholipid species in addition to PS). The authors should consider measuring DA-Raf1 binding to PS vs non-PS liposomes to confirm PS-specificity.

Minor comments

1. It appears that expression constructs for ARAF and BRAF are mouse, whereas CRAF is human (Fig. 2A). While I am not suggesting that all experiments should be repeated with human versions, this discrepancy should be made clear in the text as there are sequence differences, e.g. the BRAF S348A mutation is S365A in human BRAF which makes the related sections of text confusing.

2. On page 6 the authors state "This notion is consistent with the established concept" - to which concept are they referring?

Reviewer #2 (Comments to the Authors (Required)):

This manuscript by Takano et al. characterizes details of the mechanisms by which the ARAF isoform (splice variant) DA-RAF antagonizes RAS-MAPK signaling. This group has previously reported in a series of papers that DA-RAF, which contains the RBD-CRD RAS-binding region but lacks the kinase domain, sequesters activated RAS, interfering with its signaling. The main findings in the current manuscript reveal new details about the mechanism. Because DA-RAF lacks the two sites of phosphorylation that mediate binding of a 14-3-3 dimer stabilizing an auto-inhibitory conformation of full-length RAFs, DA-RAF localizes more to the membrane, which favours its interaction with prenylated RAS. A positively charged patch on the RBD was identified and shown by mutagenesis to be involved in membrane localization via interaction with PS lipids.

The cell biology and imaging experiments are high quality and generally support the conclusions of the manuscript, with a few exceptions, discussed below. However, I feel that something important is missing in this study in that other RAF proteins interact with membranes primarily through the CRD. The authors discuss that this has not been well studied for ARAF, however the positive residues in BRAF and CRAF that interact with the membrane are largely conserved in ARAF (and thus DA-RAF). Why did the authors choose to investigate the role of the RBD but not the CRD in membrane localization of DA-RAF? The data show that RBD is involved in membrane localization (likely as a site secondary to the CRD) but lack controls to demonstrate that the proposed positively charged patch makes any specific interaction with lipids. The 3E mutation changes the net charge by -6 and only one other positively charged residue was investigated (R52L, which disrupts interaction with RAS). The two clusters discussed are not really separate (Fig. 4a), but are on the same surface and it is possible for both to engage the membrane simultaneously (Fig S2d). In complex with RAS, part of cluster 1 becomes inaccessible and K28, K66, R68 and K69 engage the membrane (<https://doi.org/10.1073/pnas.1419895112>)

Specific concerns:

1. MD simulation and lipid-binding model: K69 appears to be the key residue for membrane localization (Fig. 4b). K69E is more disruptive than a K66E/K68E double mutant (2E) and adding K69E to the double mutant (3E) is much more disruptive. However, K69 is not involved in the MD-generated lipid interface (Fig. 4E), rather it appears to be involved in an intramolecular interaction, despite being placed on the membrane in the initial structure.
2. It appears the MD used a 100% PS membrane. The inner leaflet of the PM is ~20% PS.

Minor concerns and suggestions:

3. The authors frequently refer to DA-RAF as being 'free of autoinhibition'. This doesn't make sense to me, as DA-RAF has no kinase domain to inhibit. Rather, it is incapable of adopting the autoinhibited conformation, which means the CRD is always exposed.
4. Several cryoEM structures of BRAF have been published recently. While details for ARAF may be slightly different, the principles of regulation are likely similar. In the autoinhibited conformation with 14-3-3 bound to two sites, the CRD is sequestered in the core and cannot interact with a membrane, while the RBD is partly exposed. Interpreting the model of DA-BRAF function versus full length RAFs in light of this structural information would be helpful. The cartoon in Fig. 8 could incorporate elements of this structural information about domain orientations.
5. When the authors refer to RAF and RAS, I assume this means A/B/C-RAF and H/K/N-RAS. This should be stated explicitly, while specific isoforms should be mentioned where appropriate.
6. In paragraph 2 of the intro where domain architecture is discussed, it would be helpful to refer to the schematics now shown in Fig. 2a.
7. "The CRD also interacts with Ras C-terminal farnesyl groups anchored to the PM." This statement needs a reference.
8. Paragraph 5 of the intro discusses a LOF SNP and mutation in DA-RAF. Where are these located? The SNP and mutation should also affect ARAF as they are splice variants of the same gene?
9. Last paragraph of Intro: "it is important to elucidate the molecular mechanisms of how DA-Raf performs its intrinsic dominant-negative function to Raf proteins. We addressed this unsolved issue in this study." The authors proposed the mechanism in previous publications. Here they have characterized new details about the mechanism.
10. Final paragraph of Results: "These results indicate that the stable PM association of DA-Raf, which is brought about by the predominant Ras-binding of DA-Raf, interferes with the binding of B-Raf-C-Raf heterodimer to Ras. Consequently, the stable PM association of DA-Raf prevents the ERK pathway activation." I am not sure about the causality implied here. Could one also say PM association of DA-Raf promotes Ras-binding? It seems both events are synergistic.
11. Discussion: "These findings suggest that DA-Raf does not affect the binding of PI3K to Ras, which induces Akt activation. This is probably because the sites in Ras involved in the binding of Raf and PI3K p110 catalytic subunit are partially overlapping but distinct." PI3K binds a more extensive surface of RAS than the RAF RBD, but I believe that it would not be possible for DA-RAF and PI3K to engage RAS simultaneously. Note that PI3K can be activated via RAS-independent mechanisms.
12. Figure 4 G/H: Because GST is dimeric, the affinity of GST-DA-RAF for the liposome would be enhanced by avidity of the multivalent interaction. Nevertheless, it demonstrates that 3E reduces lipid binding, but it still binds quite well, likely via the CRD.

Reviewer #3 (Comments to the Authors (Required)):

The authors claim that (1) DA-Raf does not interact with 14-3-3 β and is therefore not in an autoinhibited state, (2) DA-Raf is PM-localized despite Ras activity (GTP-loading), and (3) DA-Raf interacts with PS in the PM through an amino acid cluster in the RBD domain.

The above are for the most part supported by the data, and the article is suitable for publication; however, some revisions are necessary to strengthen the analysis. In particular, additional description of the image analysis routines is required, as the current level of detail is insufficient. Furthermore, the controls in the pull-down assays are inadequate and should be improved (especially Figures 1 and 2). Additionally, how was the expression level of the RAF and DA-Raf constructs controlled, and how do the authors know that endogenous expression levels of DA-Raf act in a dominant negative fashion?

While the authors perform an MD simulation and provide data showing that a charge reversal mutant in this amino acid cluster (K66, R68, and K69) disrupts PS hydrogen bonds, they do not consider other possible mechanisms for stable PM interactions. The CRD, which is known to mediate lipid interactions in RAF proteins, is not discussed in this context. What about possible non-RAS protein interactions?

Additionally, while the FRAP assays address some kinetic aspects of membrane association, the authors should include direct binding data, such as SPR or ITC experiments, which would be beneficial to validate the interaction quantitatively. Similarly, the RAF dimerization assays presented through BiFC could be complemented by co-immunoprecipitation experiments to confirm dimer formation and disruption by DA-Raf under endogenous conditions.

Finally, the authors do not discuss how the full-length RAF activation cycle and the off-rate of RAFs are regulated. The kinetic stability of DA-Raf at the membrane remains unclear, and further exploration of these aspects, such as the possible lack of negative feedback regulation, would provide a more comprehensive picture of DA-Raf function.

Response to Reviewers' Comments:**Reviewer #1**

This manuscript investigates the mechanisms by which DA-Raf1 exerts its dominant-negative function on the RAS-ERK signaling pathway. The authors provide evidence that DA-Raf1, a splicing isoform of ARAF, preferentially localizes to the plasma membrane due to the absence of the CR2 14-3-3 docking site and CR3 kinase domain which play key roles in RAF autoinhibition. They also propose that the plasma membrane localization of DA-Raf1 is at least in part facilitated by an interaction between a basic amino acid cluster in DA-Raf1's RAS-binding domain and phosphatidylserine in the plasma membrane. Finally, they show that the constitutive plasma membrane localization of DA-Raf allows it to more rapidly bind active RAS than full-length RAF proteins which are localized to the cytosol in their pre-signaling state. Thus, these properties allow DA-Raf1 to efficiently inhibit RAS signaling. The Endo lab have published extensively on the roles of DA-RAF1 in human development and disease and this study provides some valuable mechanistic insights into how DA-Raf1 exerts its dominant-negative function. In addition, the identification of a RBD membrane binding interface will be of broad interest to the RAS community. However, the authors should first address a number of critical issues, especially regarding the use of appropriate controls in biochemical assays:

Major comments:

1. Figures 1A shows that KRAS pulls down similar levels of DA-Raf1 as A/B/CRAF, however there are no controls to show that these are specific interactions. The same is true for figures 2B & 2H. Examples of negative controls for Figs 1A and 2H are RAS binding impaired mutations (CRAF R89L, BRAF R188L, ARAF/A-Raf1 R53L). Mutating both 14-3-3 sites to alanine, thus alleviating 14-3-3 binding to RAF, would be an appropriate control for Fig. 2B (e.g. CRAF S259A/S621A).

Following this comment, we have added data on pull-down assay using C-Raf(R89L) as a negative control for Ras binding in Fig. 1A and Fig. 2H. We have also added data on pull-down assay using C-Raf(S259A/S621A) as a negative control for 14-3-3 β binding in Fig. 2B.

2. Figure 1D shows that expression of dominant negative KRAS 17N partially mislocalizes DA-Raf1 to the cytosol, yet in the absence of any KRAS co-expression (Fig 2A) DA-Raf1 is plasma membrane localized. To what mechanism do the authors attribute this effect?

As we have shown in the manuscript, the plasma membrane (PM) localization of DA-Raf is cooperatively regulated by the interaction with PS in the PM and the binding to active Ras. Thus, endogenous Ras proteins including K-Ras, which are active in the serum-containing culture condition, seem to contribute to the PM localization of DA-Raf in Fig. 1B. When K-Ras(S17N) is expressed, it sequesters RasGEF proteins and thereby inactivates endogenous

Ras proteins. Consequently, DA-Raf may become partially localized to the cytoplasm.

3. Again, no negative controls for RAS binding to RAF are used in BiFC experiments 1G & 2I, or for 14-3-3 binding to RAF in Fig 2C. Given that BiFC creates essentially irreversible fusion of the split fluorophore such controls are especially important to show that these interactions are specific and not artifacts of the experimental approach.

Following this comment, we have added data on BiFC analysis using C-Raf(R89L) as a negative control for Ras binding in Fig. 1G, H, and Fig. 2J. We have also added data on BiFC analysis using C-Raf(S259A/S621A) as a negative control for 14-3-3 β binding in Fig. 2C and D.

4. The conclusions section contains unnecessary speculation. This section should be made concise and focus on the facts, e.g. “We have further shown here that DA-Raf prevents the binding of B-Raf and B-Raf-C-Raf dimer (probably that of A-Raf as well)” This is conjecture as ARAF dimerization was not tested. Especially given that previous studies indicate very low levels of A-RAF dimerization (Mol Cell 2013 Feb 21;49(4):751-8.).

Although it has been reported that A-Raf dimer (A-Raf–A-Raf or A-Raf–C-Raf) is formed in cells, it is likely to be applicable to certain tumor cells (e.g., Mooz et al., 2014, Sci. Signal.; Venkatanarayan et al., 2022, Cell Rep.). Thus, following this comment, we have removed “(probably that of A-Raf as well)”, and also corrected the sentence as follows: “We have further shown here that DA-Raf prevents the binding of B-Raf to active K-Ras and B-Raf–C-Raf dimer formation at the PM, even if B-Raf adopts unclosed conformation due to unphosphorylatable CR2.”

5. On page 8 the authors state: “the interaction of the cluster 2 bAAs with PS is responsible for the PM localization of DA-Raf independently of Ras activation” however, the ability of these amino acids to interact with PS is not directly tested (presumably Folch fraction I used for liposomes contains other phospholipid species in addition to PS). The authors should consider measuring DA-Raf1 binding to PS vs non-PS liposomes to confirm PS-specificity.

Following this comment, we have added data on liposome-binding assay using PE/PC liposomes and PE/PC/PS liposomes in Fig. 4I and J. The data confirm the PS-specific binding of DA-Raf.

Minor comments:

1. It appears that expression constructs for ARAF and BRAF are mouse, whereas CRAF is human (Fig. 2A). While I am not suggesting that all experiments should be repeated with human versions, this discrepancy should be made clear in the text as there are sequence differences, e.g. the BRAF S348A mutation is S365A in human BRAF which makes the related sections of text confusing.

Following this comment, we have included the information about the animal species of A/B/C-Raf cDNAs and added the corresponding amino acid numbers of human B-Raf to mouse B-Raf mutant used in this study in the Results (p. 6, line 177) and Materials and methods (p. 14, lines 450–459) sections.

2. On page 6 the authors state “This notion is consistent with the established concept” - to which concept are they referring?

Based on your feedback and the reasons below, we have revised the relevant text.

“The established concept” refers to that the binding of Raf proteins to active Ras is indispensable for their PM localization, which represents the sentence just in front of the words. Indeed, the denoted three references describe that the recruitment of Raf proteins to the PM is initiated by active Ras, which is essential for their activation.

Matallanas et al. (2011), p. 236.

Lavoie and Therrien (2015), p. 285.

Terrell and Morrison (2019), p. 6.

Reviewer #2

This manuscript by Takano et al. characterizes details of the mechanisms by which the ARAF isoform (splice variant) DA-RAF antagonizes RAS-MAPK signaling. This group has previously reported in a series of papers that DA-RAF, which contains the RBD-CRD RAS-binding region but lacks the kinase domain, sequesters activated RAS, interfering with its signaling. The main findings in the current manuscript reveal new details about the mechanism. Because DA-RAF lacks the two sites of phosphorylation that mediate binding of a 14-3-3 dimer stabilizing an auto-inhibitory conformation of full-length RAFs, DA-RAF localizes more to the membrane, which favours its interaction with prenylated RAS. A positively charged patch on the RBD was identified and shown by mutagenesis to be involved in membrane localization via interaction with PS lipids.

The cell biology and imaging experiments are high quality and generally support the conclusions of the manuscript, with a few exceptions, discussed below. However, I feel that something important is missing in this study in that other RAF proteins interact with membranes primarily through the CRD. The authors discuss that this has not been well studied for ARAF, however the positive residues in BRAF and CRAF that interact with the membrane are largely conserved in ARAF (and thus DA-RAF). Why did the authors choose to investigate the role of the RBD but not the CRD in membrane localization of DA-RAF? The data show that RBD is involved in membrane localization (likely as a site secondary to the CRD) but lack controls to demonstrate that the proposed positively charged patch makes any specific interaction with lipids. The 3E mutation changes the net charge by -6 and only one other positively charged residue was investigated (R52L, which disrupts interaction with RAS). The two clusters discussed are not really separate (Fig. 4a), but are on the same surface and it is possible for both to engage the membrane simultaneously (Fig S2d). In complex with RAS, part of cluster 1 becomes inaccessible and K28, K66, R68 and K69 engage the membrane (<https://doi.org/10.1073/pnas.1419895112>)

Both cluster 1 and 2 bAAs are on the same surface as in this comment, and we understand the possibility that the cluster 1 bAAs, particularly K28 might also be involved in the PM binding simultaneously. The reason for excluding K28 from our investigation is that, however, it interacts with active Ras (p. 7, lines 213–215), thereby making experimental evaluation difficult. Moreover, since the roles of the cluster 2 bAAs in DA-Raf and their corresponding amino acids in Raf proteins had remained obscure, we focused on the elucidation of the mechanisms of their interaction with the PM. We further reexamined the interaction of DA-Raf CRD with the PM and have included the results in the revised manuscript (Fig. 4I, J and Fig. 5A–E). The data also include controls for the lipid (PS) specificity of bAAs in DA-Raf RBD and CRD (Fig. 4I and J).

Specific concerns:

1. MD simulation and lipid-binding model: K69 appears to be the key residue for membrane

localization (Fig. 4b). K69E is more disruptive than a K66E/K68E double mutant (2E) and adding K69E to the double mutant (3E) is much more disruptive. However, K69 is not involved in the MD-generated lipid interface (Fig. 4E), rather it appears to be involved in an intramolecular interaction, despite being placed on the membrane in the initial structure.

Video 2 shows that K66, K68, and K69 do not simultaneously bind to the DOPS bilayer, but that they are repeatedly associated with and dissociated from the DOPS bilayer during the MD simulation process. However, as this comment points out, we confirm that K69 appears to be maintained without moving from the β -sheet rather than approaching PS. Therefore, we have modified the descriptions of possible roles of K66, K68, and K69 in the interaction with PS and in an intramolecular interaction in the Results (p. 8, lines 236–244) and Discussion (p. 12, lines 378–379) sections.

2. It appears the MD used a 100% PS membrane. The inner leaflet of the PM is ~20% PS.

Following this comment, we performed a liposome-binding assay using PE/PC/PS liposomes containing 20% PS and PE/PC liposomes as a control. The results are shown in Fig. 4I and J, and described in the Results section (p. 8–9, lines 248–265).

Minor concerns and suggestions:

3. The authors frequently refer to DA-RAF as being ‘free of autoinhibition’. This doesn’t make sense to me, as DA-RAF has no kinase domain to inhibit. Rather, it is incapable of adopting the autoinhibited conformation, which means the CRD is always exposed.

Following this comment, we have modified the descriptions in the relevant places in the Abstract (p. 2, lines 31–32) and Introduction (p. 4, lines 114–116) sections.

4. Several cryoEM structures of BRAF have been published recently. While details for ARAF may be slightly different, the principles of regulation are likely similar. In the autoinhibited conformation with 14-3-3 bound to two sites, the CRD is sequestered in the core and cannot interact with a membrane, while the RBD is partly exposed. Interpreting the model of DA-BRAF function versus full length RAFs in light of this structural information would be helpful. The cartoon in Fig. 8 could incorporate elements of this structural information about domain orientations.

Following this comment, we have added some descriptions of the cryo-EM structures of B/C-Raf in the Introduction section (p. 3, lines 65–68). Furthermore, we have redrawn the schematic representation in Fig. 8 to incorporate elements of the structural information about domain orientations.

5. When the authors refer to RAF and RAS, I assume this means A/B/C-RAF and H/K/N-RAS. This should be stated explicitly, while specific isoforms should be mentioned where

appropriate.

In the original and this revised manuscripts, we have stated “the small GTPase classical Ras (H-Ras, K-Ras, and N-Ras)” (p. 3, lines 50–51) and “the Raf family of Ser/Thr kinases (A-Raf, B-Raf, and C-Raf)” (p. 3, line 54). In the revised manuscript, we have further specified Ras and Raf isoforms, if it is necessary or appropriate.

6. In paragraph 2 of the intro where domain architecture is discussed, it would be helpful to refer to the schematics now shown in Fig. 2a.

Following this comment, we have inserted “(see Fig. 2A)” in the relevant place in the Introduction section.

7. “The CRD also interacts with Ras C-terminal farnesyl groups anchored to the PM.” This statement needs a reference.

We have revised this statement and included references to the statement (p. 3, lines 76–78).

8. Paragraph 5 of the intro discusses a LOF SNP and mutation in DA-RAF. Where are these located? The SNP and mutation should also affect ARAF as they are splice variants of the same gene?

We have reported the SNP (to generate R52Q) and the mutation (to generate R52W) in the DA-Raf RBD in the reference (Kanno et al., 2018) that we have cited in the paragraph 5. In this report (Kanno et al., 2018), we have shown that A-Raf with this SNP or mutation also loses its ability to bind to active K-Ras.

9. Last paragraph of Intro: “it is important to elucidate the molecular mechanisms of how DA-Raf performs its intrinsic dominant-negative function to Raf proteins. We addressed this unsolved issue in this study.” The authors proposed the mechanism in previous publications. Here they have characterized new details about the mechanism.

Following this comment, we have replaced this sentence with “We have previously elucidated the gross mechanisms of the dominant-negative function of DA-Raf. However, since the detailed mechanisms, particularly in relation to its interaction with the PM, remained unsolved, we addressed these issues in this study.”

10. Final paragraph of Results: “These results indicate that the stable PM association of DA-Raf, which is brought about by the predominant Ras-binding of DA-Raf, interferes with the binding of B-Raf-C-Raf heterodimer to Ras. Consequently, the stable PM association of DA-Raf prevents the ERK pathway activation.” I am not sure about the causality implied here. Could one also say PM association of DA-Raf promotes Ras-binding? It seems both events are synergistic.

Following this comment, we have revised the relevant sentence (p. 11, lines 337–339).

11. Discussion: “These findings suggest that DA-Raf does not affect the binding of PI3K to Ras, which induces Akt activation. This is probably because the sites in Ras involved in the binding of Raf and PI3K p110 catalytic subunit are partially overlapping but distinct.” PI3K binds a more extensive surface of RAS than the RAF RBD, but I believe that it would not be possible for DA-RAF and PI3K to engage RAS simultaneously. Note that PI3K can be activated via RAS-independent mechanisms.

Following this comment, we have revised the relevant statements (p. 13, lines 423–429).

12. Figure 4 G/H: Because GST is dimeric, the affinity of GST-DA-RAF for the liposome would be enhanced by avidity of the multivalent interaction. Nevertheless, it demonstrates that 3E reduces lipid binding, but it still binds quite well, likely via the CRD.

We newly examined the interaction between DA-Raf and PS by a liposome-binding assay using CRD and RBD mutants. The results imply that DA-Raf CRD bAAs, as well as DA-Raf RBD cluster 2 bAAs, are involved in the PS binding (Fig. 4I and J; p. 8–9, lines 248–265). This PS-binding ability of CRD seems to be the reason that DA-Raf(3E) was coprecipitated with PS-containing liposomes and PE/PC/PS liposomes at a low level but still to some degree.

Reviewer #3

The authors claim that (1) DA-Raf does not interact with 14-3-3 β and is therefore not in an autoinhibited state, (2) DA-Raf is PM-localized despite Ras activity (GTP-loading), and (3) DA-Raf interacts with PS in the PM through an amino acid cluster in the RBD domain.

The above are for the most part supported by the data, and the article is suitable for publication; however, some revisions are necessary to strengthen the analysis. In particular, 1) additional description of the image analysis routines is required, as the current level of detail is insufficient. Furthermore, 2) the controls in the pull-down assays are inadequate and should be improved (especially Figures 1 and 2). Additionally, 3) how was the expression level of the RAF and DA-Raf constructs controlled, and 4) how do the authors know that endogenous expression levels of DA-Raf act in a dominant negative fashion?

- 1) Following this comment, we have corrected or added descriptions of the image analyses in the Materials and methods section (pp. 17–18).
- 2) We have added data on pull-down assay using C-Raf(R89L) as a negative control for Ras binding in Fig. 1 A and Fig. 2 H. We have also added data on pull-down assay using C-Raf(S259A/S621A) as a negative control for 14-3-3 β binding in Fig. 2 B. Moreover, we have added data on BiFC analysis using C-Raf(R89L) as a negative control for Ras binding in Fig. 1 G and H, and Fig. 2 I and J. We have also added data on BiFC analysis using C-Raf(S259A/S621A) as a negative control for 14-3-3 β binding in Fig. 2 C and D.
- 3) We used almost equimolar amounts of plasmid constructs of Raf proteins and DA-Raf for transfection to adjust their expression levels (p. 15, lines 475–477).
- 4) Our previous studies show that the endogenous expression levels of DA-Raf act in a dominant-negative manner to Raf proteins. These studies include the experiments with DA-Raf knockdown cells (Yokoyama et al., 2007) and knockout mice (Watanabe-Takano et al., 2014).

While the authors perform an MD simulation and provide data showing that a charge reversal mutant in this amino acid cluster (K66, R68, and K69) disrupts PS hydrogen bonds, they do not consider other possible mechanisms for stable PM interactions. 1) The CRD, which is known to mediate lipid interactions in RAF proteins, is not discussed in this context. 2) What about possible non-RAS protein interactions?

- 1) We newly examined the interaction of DA-Raf CRD with PS, by using CRD mutants together with the RBD mutants in a liposome-binding assay. The results imply that DA-Raf CRD bAAs, as well as DA-Raf RBD cluster 2 bAAs, are involved in the PS binding. The results are shown in Fig. 4I, J, described in the Results section (p. 8–9, lines 248–265), and discussed in the Discussion section (p. 12–13, lines 383–403).
- 2) Interactions of DA-Raf with non-Ras proteins may be important points to explore the mechanisms of DA-Raf functions. However, it seems to be beyond the subject of this study. We would like to examine these matters in next studies.

Additionally, while the FRAP assays address some kinetic aspects of membrane association, 1) the authors should include direct binding data, such as SPR or ITC experiments, which would be beneficial to validate the interaction quantitatively. Similarly, 2) the RAF dimerization assays presented through BiFC could be complemented by co-immunoprecipitation experiments to confirm dimer formation and disruption by DA-Raf under endogenous conditions.

1) The SPR or ITC analysis may quantitatively validate the association of K-Ras–Raf/DA-Raf with the PM. However, we are unable to conduct these analyses due to the limitation of facilities.

2) B-Raf(S348A)–C-Raf dimerization and its disruption by DA-Raf, which are analyzed by BiFC analysis, may be complemented by coimmunoprecipitation assay. However, the results of the coimmunoprecipitation analysis may complicate their interpretation due to the transitional cellular localization of Raf proteins. In contrast, the results of BiFC analysis present the information about both the interaction and the localization at a time.

Finally, the authors do not discuss how the full-length RAF activation cycle and the off-rate of RAFs are regulated. The kinetic stability of DA-Raf at the membrane remains unclear, and further exploration of these aspects, such as the possible lack of negative feedback regulation, would provide a more comprehensive picture of DA-Raf function.

These are important points to comprehensively reveal the dominant-negative function of DA-Raf in vivo, and we would like to examine these matters in next studies. Thus, we added these aspects in the Discussion section (p. 13, lines 408–412).

September 8, 2025

Re: Life Science Alliance manuscript #LSA-2025-03300-TR

Dr. Kazunori Takano
Chiba University
Department of Biology, Graduate School of Science
1-33, Yayoi-cho, Inage-ku
Chiba, Chiba 263-8522
Japan

Dear Dr. Takano,

Thank you for submitting your revised manuscript entitled "DA-Raf dominant-negatively antagonizes Raf by its predominant binding to the plasma membrane and Ras" to Life Science Alliance. The manuscript has been seen by all of the original reviewers whose comments are appended below. While the reviewers continue to be overall positive about the work in terms of its suitability for Life Science Alliance, a few important issues remain. As you will note, Reviewer 2 has reiterated three inconsistencies between the experimental data and the model. We concur with this reviewer's suggestions and you must address these points at least in the discussion. Next, Reviewer 3 has suggested the need for a thorough grammar check, which we agree must be done. If you prefer to use AI-based spell/grammar checkers, kindly do so. Please specify in the methods if AI-based spell/grammar checks were used in the manuscript.

Our general policy is that papers are considered through only one revision cycle; however, given that the suggested changes are relatively minor, we are open to one additional short round of revision.

Please submit the final revision along with a letter that includes a point by point response to the remaining reviewer comments.

To upload the revised version of your manuscript, please log in to your account: <https://lsa.msubmit.net/cgi-bin/main.plex>
You will be guided to complete the submission of your revised manuscript and to fill in all necessary information.

B. MANUSCRIPT ORGANIZATION AND FORMATTING:

Sincerely,

Sarita Hebbar, PhD
Scientific Editor
Life Science Alliance
<http://www.lsjournal.org>

Reviewer #1 (Comments to the Authors (Required)):

The authors have adequately addressed all of my comments and critiques.

Reviewer #2 (Comments to the Authors (Required)):

The authors have addressed most of my concerns in this revision, however there is still an inconsistency between the experimental data and the MD model of the DA-RAF membrane interface, which should be addressed (or at least discussed) before publication.

Re: # 1.

K69 is a key residue for membrane localization, since K69E is more disruptive than a K66E/R68E double mutant (2E). The MD model, in which K69 forms an intramolecular interaction and is not involved in membrane interaction, is thus not consistent with the experimental data.

The structure used for the MD model is an NMR ensemble of 20 structures (1WXM). The K69 sidechain exhibits some flexibility in this ensemble and is only involved in intramolecular interactions in a few of the ensemble conformers, for example when it approaches E89. The authors may find a resolution to the above inconsistency by looking at the NMR ensemble as a whole. Assuming the input for the MD was a conformer with a K69 intramolecular interaction, it appears this was preserved for the duration of the simulation, but a conformer in which the K69 sidechain is exposed might produce a different result.

Re: # 2.

"It appears the MD used a 100% PS membrane."

The response to Comment #2 refers to the liposome-binding assay, but the question was about the molecular dynamics simulations. If the authors perform another MD simulation to address Comment #1, they might consider using a more physiological (~20% PS) membrane, otherwise they should include a caveat that this PS concentration is much higher than the plasma membrane.

Minor:

Re: # 8.

The LOF SNP and mutation in DA-RAF affect the same site (R52) as a designed mutation (R52L) that was used extensively, thus I encourage the authors to tie these together in the manuscript.

Reviewer #3 (Comments to the Authors (Required)):

I have read the authors' responses to my comments and reviewed the revisions made to the manuscript. I acknowledge that certain limitations of the facility make it difficult to perform biophysical experiments such as SPR, and that some of my comments may be addressed in future work. Overall, the study is suitable for publication. However, there are minor grammatical errors in the revised manuscript that the authors should correct.

Response to Reviewers' Comments:**Reviewer #1**

The authors have adequately addressed all of my comments and critiques.

Reviewer #2

The authors have addressed most of my concerns in this revision, however there is still an inconsistency between the experimental data and the MD model of the DA-RAF membrane interface, which should be addressed (or at least discussed) before publication.

Re: # 1.

K69 is a key residue for membrane localization, since K69E is more disruptive than a K66E/R68E double mutant (2E). The MD model, in which K69 forms an intramolecular interaction and is not involved in membrane interaction, is thus not consistent with the experimental data.

The structure used for the MD model is an NMR ensemble of 20 structures (1WXM). The K69 sidechain exhibits some flexibility in this ensemble and is only involved in intramolecular interactions in a few of the ensemble conformers, for example when it approaches E89. The authors may find a resolution to the above inconsistency by looking at the NMR ensemble as a whole. Assuming the input for the MD was a conformer with a K69 intramolecular interaction, it appears this was preserved for the duration of the simulation, but a conformer in which the K69 sidechain is exposed might produce a different result.

Following these comments, we have discussed particularly the role of DA-Raf K69 in the MD simulation model and in the PM localization analysis (p. 12). We have also included the discussion about the importance of MD simulations with active Ras for the elucidation of the intracellular interaction of DA-Raf with PS in the PM. These discussions may be useful to resolve the seeming inconsistency between the MD simulation model and the data in the PM localization analysis.

Re: # 2.

"It appears the MD used a 100% PS membrane."

The response to Comment #2 refers to the liposome-binding assay, but the question was about the molecular dynamics simulations. If the authors perform another MD simulation to address Comment #1, they might consider using a more physiological (~20% PS) membrane, otherwise they should include a caveat that this PS concentration is much higher than the plasma membrane.

Following this comment, we have revised the relevant places in the Results section (p. 8), the Discussion section (p. 12) and the Materials and Methods section (p. 18) to clarify that the MD simulations applied a 100% PS membrane and that the inner leaflet of the PM is composed of ~20% PS.

Minor:

Re: # 8.

The LOF SNP and mutation in DA-RAF affect the same site (R52) as a designed mutation (R52L) that was used extensively, thus I encourage the authors to tie these together in the manuscript.

Following this comment, we have stated explicitly that the SNP R52Q and R52W mutant in a cancer, as well as R52L mutant, are incompetent to suppress the K-Ras-induced transformation in the Introduction section (p. 4).

Reviewer #3

I have read the authors' responses to my comments and reviewed the revisions made to the manuscript. I acknowledge that certain limitations of the facility make it difficult to perform biophysical experiments such as SPR, and that some of my comments may be addressed in future work. Overall, the study is suitable for publication. However, there are minor grammatical errors in the revised manuscript that the authors should correct.

Following this comment, we have proofread the text and corrected grammatical errors by using Grammarly EDU. This is stated in the Materials and methods section (pp. 19–20).

September 19, 2025

RE: Life Science Alliance Manuscript #LSA-2025-03300-TRR

Dr. Kazunori Takano
Chiba University
Department of Biology, Graduate School of Science
1-33, Yayoi-cho, Inage-ku
Chiba, Chiba 263-8522
Japan

Dear Dr. Takano,

Thank you for submitting your revised manuscript, entitled "DA-Raf dominant-negatively antagonizes Raf by its predominant binding to the plasma membrane and Ras". We acknowledge your second resubmission with the inclusion of minor revisions.

We would be happy to publish your paper in Life Science Alliance pending final revisions necessary to meet our formatting guidelines.

- For the "Data Availability" section, kindly provide (source data) DOI/link that is functional.
- Please provide details for imaging-FRAP experiments (name of microscope, n.a., and magnification of objective, temperature of stage).
- Please rephrase the sentence in the abstract" Accordingly, DA-Raf dominant-negatively antagonizes the Ras-ERK pathway."
- We encourage you to make the title clearer, "DA-Raf dominant-negatively antagonizes Raf by its predominant binding to the plasma membrane and Ras".
- Please use the [10 author names, et al.] format in your references (i.e., limit the author names to the first 10)
- The contributions selected for Toshiki Itoh do not qualify them for authorship. Please either update the contributions in our system and the Author Contributions section of the manuscript, or let us know if the author needs to be removed (and added eventually to the acknowledgment section)
- please add a callout for Figure 3B-D; 8A-D; S1A-C and S2A-D to your main manuscript text;
- Please upload your Table in editable .doc or Excel format
- Please upload a clean manuscript file without the colored text
- Please upload all figure files as individual ones, including the supplementary figure files; all figure legends should only appear in the main manuscript file
- Please add the X and Bluesky handles of your host institute/organization, as well as your own and/or one of the authors in our system
- Please be sure that the authorship listing and order is correct

LSA now encourages authors to provide a 30-60 second video where the study is briefly explained. We will use these videos on social media to promote the published paper and the presenting author (for examples, see <https://docs.google.com/document/d/1-UWCfbE4pGcDdcgzcmiuJl2XMBJnxKYeqRvLLrLSo8s/edit?usp=sharing>). Corresponding or first-authors are welcome to submit the video. Please submit only one video per manuscript. The video can be emailed to contact@life-science-alliance.org

A. FINAL FILES:

- An editable version of the final text (.DOC or .DOCX) is needed for copyediting (no PDFs).
- High-resolution figure, supplementary figure and video files uploaded as individual files: See our detailed guidelines for

preparing your production-ready images, <https://www.life-science-alliance.org/authors>

B. MANUSCRIPT ORGANIZATION AND FORMATTING:

Thank you for your attention to these final processing requirements. Please revise and format the manuscript and upload materials as soon as you are able.

Sincerely,

Sarita Hebbar, PhD
Scientific Editor
Life Science Alliance
<http://www.lsjournal.org>

September 25, 2025

RE: Life Science Alliance Manuscript #LSA-2025-03300-TRRR

Dr. Kazunori Takano
Chiba University
Department of Biology, Graduate School of Science
1-33, Yayoi-cho, Inage-ku
Chiba, Chiba 263-8522
Japan

Dear Dr. Takano,

Thank you for submitting your Research Article entitled "DA-Raf dominant-negatively regulates Raf by preferentially binding to the plasma membrane and Ras". It is a pleasure to let you know that your manuscript is now accepted for publication in Life Science Alliance. Congratulations on this interesting work.

Your manuscript will now progress through copyediting and proofing. At the proofs stage, we encourage you to phrase the title differently. A suggested example is "DA-Raf is a dominant negative regulator of Raf proteins by plasma membrane association and predominating the binding to Ras".

It is journal policy that authors provide original data upon request. Reviews, decision letters, and point-by-point responses associated with peer-review at Life Science Alliance will be published, alongside the manuscript. If you do want to opt out of having the reviewer reports and your point-by-point responses displayed, please let us know immediately.

DISTRIBUTION OF MATERIALS:

Again, congratulations on a very nice paper. I hope you found the review process to be constructive and are pleased with how the manuscript was handled editorially. We look forward to future exciting submissions from your lab.

Sincerely,

Sarita Hebbar, PhD
Scientific Editor
Life Science Alliance
<http://www.lsajournal.org>